REPORT

# DBF4, not DRF1, is the crucial regulator of CDC7 kinase at replication forks

Anja Göder[1], Chrystelle Antoinat Maric[2], Michael D. Rainey[1], Aisling O'Connor[1], Chiara Cazzaniga[1], Daniel Shamavu[1], Jean-Charles Cadoret[2], and Corrado Santocanale[1]

CDC7 kinase is crucial for DNA replication initiation and is involved in fork processing and replication stress response. Human CDC7 requires the binding of either DBF4 or DRF1 for its activity. However, it is unclear whether the two regulatory subunits target CDC7 to a specific set of substrates, thus having different biological functions, or if they act redundantly. Using genome editing technology, we generated isogenic cell lines deficient in either DBF4 or DRF1: these cells are viable but present signs of genomic instability, indicating that both can independently support CDC7 for bulk DNA replication. Nonetheless, DBF4-deficient cells show altered replication efficiency, partial deficiency in MCM helicase phosphorylation, and alterations in the replication timing of discrete genomic regions. Notably, we find that CDC7 function at replication forks is entirely dependent on DBF4 and not on DRF1. Thus, DBF4 is the primary regulator of CDC7 activity, mediating most of its functions in unperturbed DNA replication and upon replication interference.

## Introduction

In eukaryotic cells, DNA replication follows a highly regulated program to ensure the faithful and complete duplication of the genome (Costa and Diffley, 2022; Moiseeva and Bakkenist, 2018).

The CDC7 Ser/Thr kinase phosphorylates multiple subunits of the minichromosome maintenance (MCM) complex at replication origins allowing the recruitment of additional co-factors such as CDC45 and GINS and the activation of the helicase (Masai et al., 2006; Sheu and Stillman, 2010; Zou and Stillman, 2000). CDC7-dependent phosphorylation of the MCM complex is counteracted by RIF1-PP1 phosphatase. Thus, the efficiency of replication initiation is determined by the interplay between kinases and phosphatases, contributing to the well-defined spatial–temporal coordination of origin firing (Alver et al., 2017; Hiraga et al., 2017).

In addition, CDC7 is involved in translesion DNA synthesis (Day et al., 2010; Lee et al., 2012; Yamada et al., 2013) and replication stress response where, by phosphorylating CLASPIN, it facilitates CHK1 activation by the Ataxia telangiectasia and Rad3 related (ATR) kinase (Kim et al., 2008; Rainey et al., 2013; Yang et al., 2019). The ATR-CHK1 pathway is then responsible for cellular responses to replication stress, including transcriptional reprogramming and withdrawal from the cell cycle. Critically, the ATR-CHK1 pathway limits DNA damage and genome instability by stabilizing stalled replication forks and preventing further origin firing (Bass and Cortez, 2019; Wagner et al., 2016). Recently, we demonstrated that hCDC7 kinase regulates MRE11 nuclease, thus promoting fork restart and modulating fork speed. CDC7 associates with active and stalled replisomes, where it likely phosphorylates key substrates, possibly including MRE11 itself, as well as other proteins involved in homologous recombination DNA repair (Iwai et al., 2021; Jones et al., 2021; Rainey et al., 2020b).

In most organisms, CDC7 kinase activity is fully dependent upon its interaction with DBF4, which was first identified in budding yeast (Johnston and Thomas, 1982). DBF4 is evolutionarily conserved although sequence homology is restricted to three motifs, named N, M, and C motifs, for their relative position in the protein. The M and C motifs interact with the N and C lobes of CDC7, respectively, and stabilize the kinase in an active conformation, while the N motif contains a BRCA1 C-terminus–like domain and mediates protein–protein interactions (Cheng et al., 2022; Greiwe et al., 2022; Hughes et al., 2012; Ogino et al., 2001; Saleh et al., 2022).

In human cells, two DBF4-like proteins have been described: DBF4 and DBF4B, also called DRF1 (DBF4 related factor 1) or ASKL1 (Montagnoli et al., 2002; Yoshizawa-Sugata et al., 2005); for clarity, we will refer to it as DRF1. Both DBF4 and DRF1 form stable complexes with CDC7 (Tenca et al., 2007), and in proliferating cells, these are expressed almost simultaneously. In

[1]Centre for Chromosome Biology, School of Biological and Chemical Sciences, University of Galway, Galway, Ireland; [2]Université Paris Cité, CNRS, Institut Jacques Monod, Paris, France.

Correspondence to Corrado Santocanale: corrado.santocanale@universityofgalway.ie.

other species, however, their pattern of expression can vary greatly. As an example, in *Xenopus laevis*, DRF1 is only expressed during early embryonic development and is essential for DNA synthesis, and at later stages is replaced by DBF4 (Silva et al., 2006; Takahashi and Walter, 2005).

Despite the existence of two regulatory subunits for human CDC7 being known for over 20 years (Montagnoli et al., 2002), our understanding of DRF1 and DBF4's functions has been limited due to technical challenges including (1) difficulties in the detection of endogenous proteins and (2) the limited efficiency of siRNA in reducing the DRF1 mRNA levels.

Here, we report DBF4 as the primary mediator of CDC7 activity during replication stress, where it contributes to fork processing and checkpoint signaling directly at stalled replication forks.

## Results and discussion

### CDC7 activity is primarily mediated by DBF4 and only to a lesser extent by DRF1

To dissect the roles of CDC7's regulatory subunits, we generated DBF4- and DRF1-deficient cells by transfecting MCF10A EditR cells (Rainey et al., 2017), stably expressing Cas9, with plasmids expressing short guide RNAs (sgRNAs) either targeting exon 3 of *DBF4* or exon 9 of *DRF1* (Fig. 1, A and B). From the pools, two DBF4 and two DRF1 independent clones were isolated and characterized: DBF4 clone 11 and 30 (DBF4-11 and DBF4-30) displayed homozygous deletions of 13 and 5 nucleotides, respectively, DRF1 clone 5 (DRF1-5) a homozygous four nucleotide deletion, and DRF1 clone 7 (DRF1-7) a single nucleotide insertion into exon 9 of *DRF1* (Fig. S1, A and E). In all cases, gene editing resulted in a premature stop codon, generating truncated proteins lacking the critical domains required for the binding and activation of CDC7. Thus, these could be considered a bona fide loss of function with respect to CDC7 activation. Using an antibody generated against the C-terminus of DBF4, we observed an immunoreactive band at ~110 kDa, which is consistent with DBF4 previously reported migration in SDS-PAGE (Montagnoli et al., 2002). This band is missing in both DBF4-11 and DBF4-30 clones; unexpectedly, a new band of ~55 kDa was detected only in DBF4-30 cells. We reckon that this polypeptide is produced by translation from an internal start site, thus lacking the N and most of the M motif (Fig. S1, B–D). As we failed to detect endogenous DRF1 by western blotting, we used an anti-DRF1 antibody to co-immunoprecipitate CDC7, which was detected when extracts were prepared from parental but not from DRF1-5 and DRF1-7 cells, indicating that DRF1 expression is compromised in these clones (Fig. S1, F and G). As a note of caution, we cannot exclude the possibility that either through exon skipping events or aberrant translation, low levels of proteins with partial functionality may be present; thus, throughout this work, we define these cell lines as DBF4- and DRF1-deficient and not knockout cells. Unless otherwise indicated, experiments were preferentially performed with MCF10A DBF4-11 and DRF1-7 cells.

Viable DBF4- and DRF1-deficient cells indicate that either DBF4 or DRF1 can support CDC7's essential function in cell proliferation. Our attempts to knock out both *DBF4* and *DRF1* were unsuccessful, as we were unable to recover viable clones, like our previous attempts at generating a CDC7 knockout (Rainey et al., 2017). Thus, unlike CDC7, neither DBF4 nor DRF1 is essential in MCF10A cells, leading us to analyze the datasets from CRISPR/Cas9 screens assessing the dependency of cell lines on a given gene for proliferation (DepMap, 2023; Meyers et al., 2017). All the 1,095 cell lines tested showed a very strong dependency on *CDC7*, therefore defining *CDC7* as a common essential gene. Most cell lines displayed some level of dependency on *DBF4*, but this was less marked than with CDC7, while only 13 of 1,095 cell lines showed a dependency on *DRF1* (Fig. 1 C).

To assess DBF4 and DRF1 contributions to CDC7 activity in a cellular context, we analyzed S40/41 MCM2 phosphorylation, a well-established CDC7 substrate (Montagnoli et al., 2006); this was nearly abolished in MCF10A EditR by treatment with the CDC7i XL413, drastically reduced in DBF4-deficient cells but only partially affected in the DRF1-7 clone (Fig. 1 D). As S40 MCM2 phosphorylation is cell cycle regulated and dependent on a priming kinase phosphorylating S41 (Montagnoli et al., 2006), we also looked at CDC7 in mitotic cells, when it is highly modified partially through autophosphorylation, resulting in an electrophoretic mobility shift in SDS-PAGE (Jiang et al., 1999; Knockleby et al., 2016). Upon nocodazole treatment, a fraction of CDC7 molecules migrated slower in MCF10A EditR cells, the shift was strongly attenuated by XL413 and reduced by the loss of DBF4 and of DRF1, albeit to a lesser extent (Fig. 1 E). CDC7 inhibition is associated with irregular progression through mitosis, often resulting in the formation of micronucleated cells (Cazzaniga et al., 2024; Martin et al., 2022). Interestingly, while we did not observe a significant change in the percentage of micronucleated cells in DBF4-deficient cells, these clearly accumulated in DRF1-deficient cells (Fig. 1, F and G), which could be due to minor impairment of DNA replication/repair or defective chromosome segregation. A tempting hypothesis is that DRF1 may modulate the timing of abscission at the end of the cell cycle, a non-essential process in which CDC7 was recently shown to be involved and that, if impaired, can lead to micronucleated cells (Luessing et al., 2022).

### DBF4 drives CDC7 in DNA replication

To assess DBF4 and DRF1 contribution to DNA replication, parental, DBF4-, and DRF1-deficient cells were labeled for 30 min with the thymidine analog EdU and analyzed by flow cytometry. DNA synthesis was only partially reduced in DBF4-deficient cells, more evidently in cells in mid- to late S-phase, but importantly, it was not reduced in DRF1-deficient cells (Fig. 2, A and B). Consistently, siRNA targeting of DBF4 reduced the rate of DNA synthesis, while DRF1-targeting siRNAs did not (Fig. S2, A–D). By targeting both subunits in parental cells with siRNA, we did not observe a further reduction in EdU incorporation compared with siDBF4 alone (Fig. S2, C and D). This result was confirmed by the lack of reduction in the rate of DNA synthesis when transfecting DRF1 siRNAs into DBF4-deficient cells, while targeting DBF4 in MCF10A DRF1-deficient cells reduced but did not abolish EdU incorporation (Fig. S2, D and E). Downregulation of DBF4 and DRF1 mRNAs was monitored by

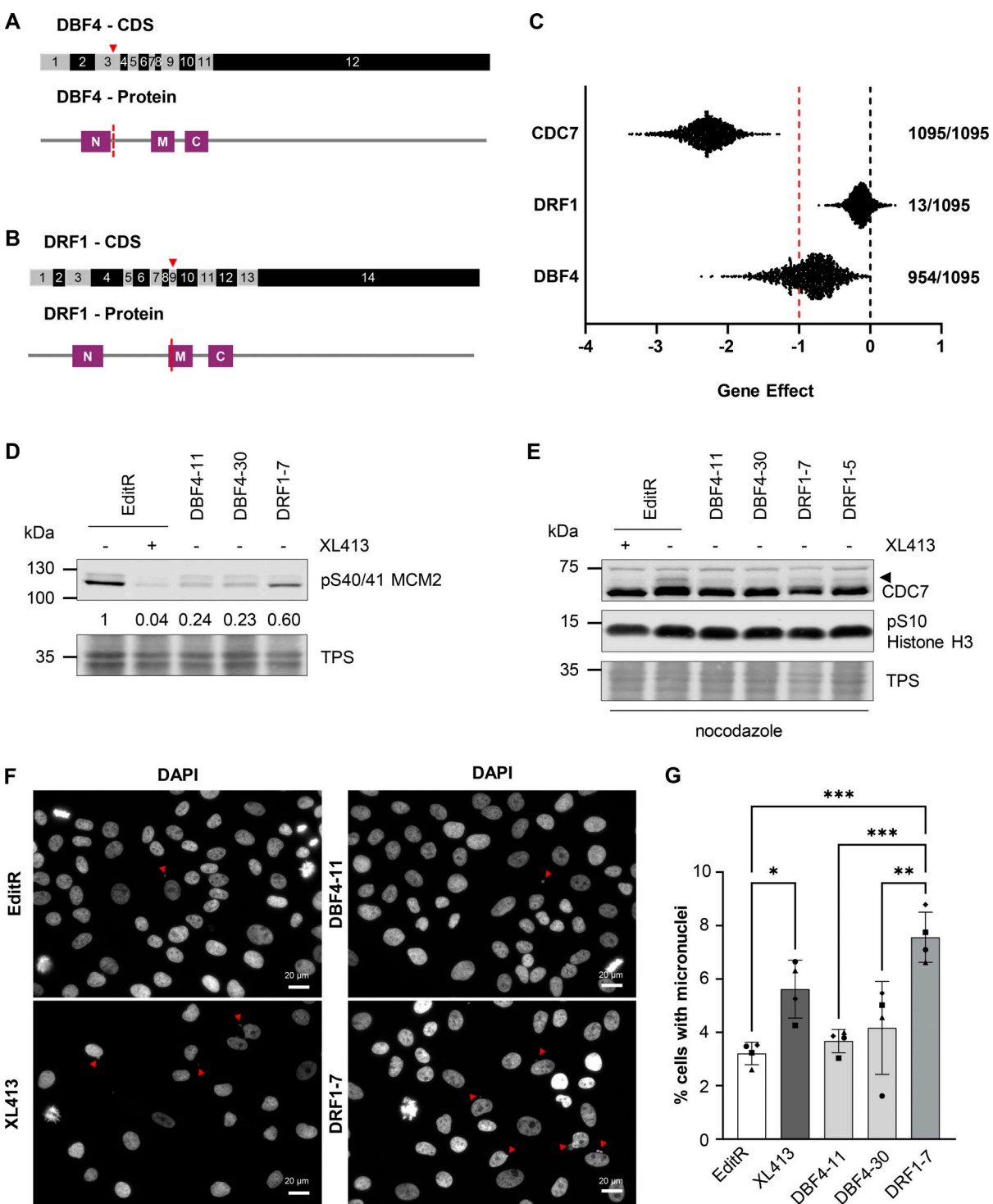

Figure 1. **CDC7 activity is primarily mediated by DBF4 and only to a minimal extent by DRF1. (A and B)** Schematic representation of gene editing approach for the generation of DBF4- (A) and DRF1-deficient (B) cells. Red triangle marks the position of the Cas9 cut site in the coding sequence (CDS) for DBF4 or DRF1, exons are numbered. M, N, and C motifs in both proteins are aligned with the CDS and marks indicate the position of the Cas9 cut site relative to the protein sequence. **(C)** Comparison of CHRONOS dependency scores (gene effect) obtained in CRISPR/Cas9 knockout screens for *CDC7*, *DBF4*, and *DRF1* available from DepMap portal. A score of 0 (black line) or higher would describe a non-essential gene, a score of −1 (red line) corresponds to the median of all pan-essential genes. A lower score describes a higher chance of the gene of interest being essential. Numbers represent the number of cell lines, which have been classified as dependent on *CDC7*, *DBF4*, or *DRF1* of 1,095 tested cell lines. **(D)** MCF10A EditR, MCF10A DBF4-11 and -30, and MCF10A DRF1-7 were treated with 10 µM XL413 or DMSO for 24 h. Whole-cell extracts were prepared and analyzed by immunoblotting with indicated antibodies. TPS was used as loading control. Numbers indicate relative changes in MCM2 phosphorylation compared with the DMSO-treated control cell line and normalized to TPS in the displayed blot. Data are representative of at least three independent experiments. **(E)** MCF10A EditR, MCF10A DBF4-11 and -30, and MCF10A DRF1-5 and -7 were treated with 10 µM XL413 or DMSO for 24 h. 16 h before harvesting, cells were additionally treated with 0.2 µg/ml nocodazole. Whole-cell extracts were

prepared and analyzed by immunoblotting with indicated antibodies. TPS was used as loading control. Triangle marks the mobility shift of CDC7. Data are representative of three independent experiments. **(F)** MCF10A EditR were either mock or treated with 10 µM XL413 for 24 h. MCF10A DBF4-11 and -30 and MCF10A DRF1-7 were only mock-treated. Cells were fixed, stained with DAPI to visualize DNA, and analyzed by fluorescence microscopy. Representative images of four independent experiments are shown (scale bar, 20 µm). Red triangles indicate micronuclei. The brightness of images was adjusted for all samples to aid visualization. **(G)** Graph shows the percentage of cells with micronuclei for four independent experiments, mean ± SD. At least 275 cells were analyzed per condition for each experiment. Statistical analysis was performed using one-way ANOVA with Tukey's multiple comparison test (*P < 0.05, **P < 0.01, ***P < 0.001). Source data are available for this figure: SourceData F1.

quantitative PCR (qPCR), and while DBF4 siRNAs efficiently downregulated DBF4 expression, the efficiency of DRF1 siRNA was limited to ~60% with residual expression likely masking relevant phenotypes (Fig. S2, A and B). Since these experiments did not reveal an obvious role for DRF1 in replication, we further

investigated replication dynamics in DBF4-deficient cells by DNA combing assay. We found that in DBF4-deficient cells, the average replication fork speed was increased from 1.2 kb/min to 1.7 kb/min (Fig. 2, C and D), a phenotype recapitulated by treatment with XL413 (Fig. 2, C and D) and previously reported

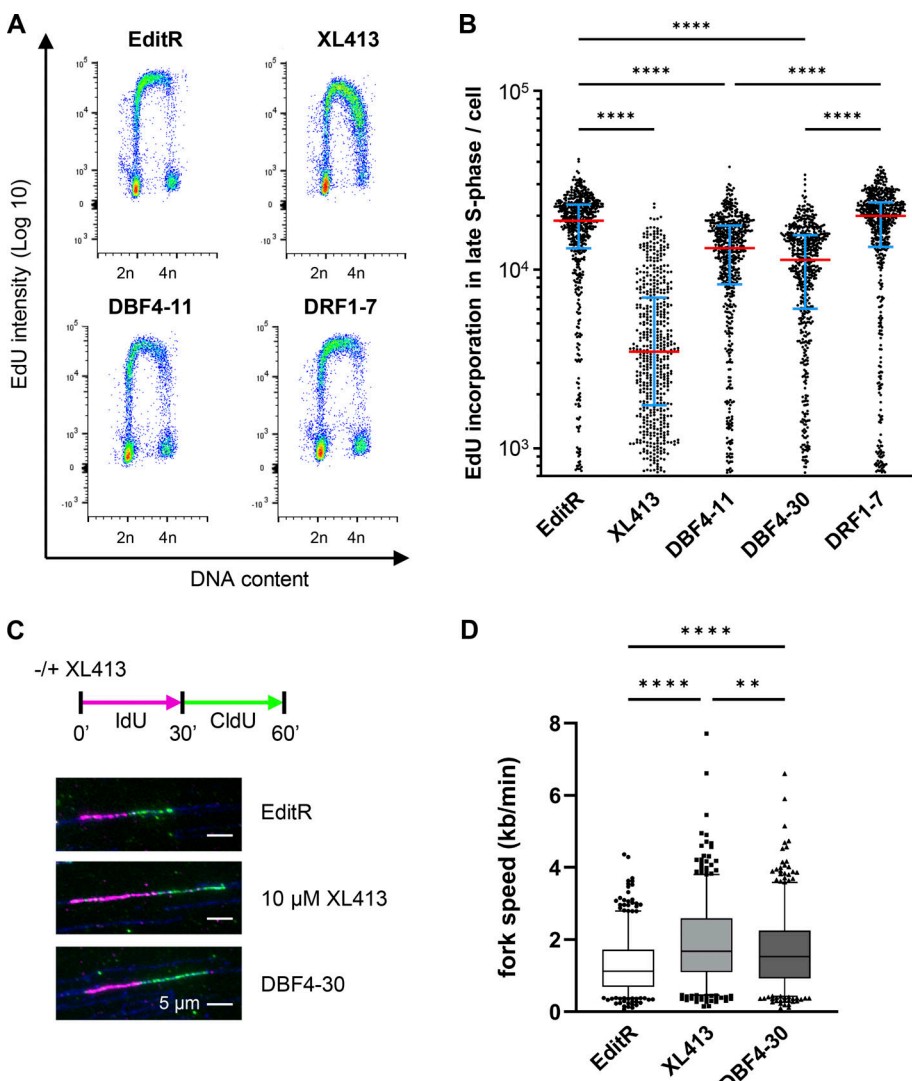

Figure 2. **DBF4, not DRF1, is the major contributor to CDC7 activity in DNA replication. (A)** MCF10A EditR, MCF10A DBF4-11 and -30, and MCF10A DRF1-7 were treated with 10 µM XL413 or DMSO for 24 h. For flow cytometry analysis, cells were labeled with 10 µM EdU 30 min prior to harvest. Representative images from one of three independent experiments are shown. **(B)** Fluorescence intensity, proportional to EdU incorporation, in late S-phase cells for representative experiment displayed in A. Red lines indicate the medians and blue lines show the interquartile range extending from the 25th to the 75th percentile of 552 cells. Statistical analysis was performed using one-way ANOVA with Tukey's multiple comparison test (****P < 0.0001). **(C)** MCF10A EditR and MCF10A DBF4-30 were treated with 10 µM XL413 or DMSO for 24 h and then labeled with IdU (magenta) for 30 min. IdU was washed off and cells were labeled with CldU (green) for 30 min in the presence of 10 µM XL413 or DMSO. Representative fibers for each treatment/cell line are shown. **(D)** Analysis of replication fork speed in the experiment described in C. At least 100 tracks were analyzed for each condition of three independent experiments. Box plots display the median and 5–95% range. Statistical analysis was performed using one-way ANOVA with Tukey's multiple comparison test (**P < 0.01, ****P < 0.0001).

with other CDC7 inhibitors (Iwai et al., 2019; Montagnoli et al., 2008).

DBF4 loss reduced the rate of DNA synthesis, but increased replication fork speed, an effect generally attributed to a compensatory mechanism responding to a reduction in origin firing (Zhong et al., 2013) and more recently also to differential fork processing (Merchut-Maya et al., 2019; Rainey et al., 2020b).

Unexpectedly, despite the basic redundancy between DBF4 and DRF1, we did not detect synergy in reducing the rate of DNA replication when targeting both DBF4 and DRF1 by multiple approaches. The lack of synergy could be of biological or technical nature such as (1) residual low levels of DBF4/DRF1 proteins remaining in the cells, (2) alternative yet unidentified mechanisms of kinase activation, and (3) minimal residual enzymatic activity of hCDC7 kinase not requiring an activating subunit. Further work will be required to test these hypotheses.

## DBF4 loss changes global replication timing (RT) like CDC7 inhibition

We then performed RT experiments in MCF10A EditR with or without CDC7i treatment and in the DBF4-deficient cells. Cells were labeled with a short pulse of BrdU and divided into early and late S-phase fractions. The DNA of neosynthesis in these two fractions was hybridized on whole-genome microarrays, thus generating differential RT profiles as previously described (Hadjadj et al., 2016, 2020). XL413-treated MCF10A EditR and DBF4-deficient cells showed variations in the RT at 322 (125 advanced and 197 delayed) and 185 (58 advanced and 127 delayed) regions, respectively, mostly distributed in clusters of several adjacent advanced or delayed regions (Fig. 3, A and B; and Fig. S3, A and B). In XL413-treated cells, RT is altered in 20.6% of the genome, whereas 13.7% of the RT is changed in DBF4-deficient cells. In line with DBF4 being required for most of CDC7's activity, ~70% of the regions changing their RT in DBF4-deficient cells were similarly affected by XL413 (Fig. 3 C). RT advancement in XL413-treated cells and DBF4-deficient cells mainly occurred in early replicating regions; however, some late replicating regions and timing transition regions (TTR) also displayed earlier timing (Fig. 3, A, B, D, and E; and Fig. S3 A). Similarly, delayed regions were primarily found in parts of the genome replicating in early S-phase or TTR regions (Fig. 3, A, B, D, and E; and Fig. S3 B). Further analysis revealed that advanced regions are enriched in CpG islands and putative G4 which are often enriched at replication origins (Picard et al., 2014) as well as constitutive origins (Fig. S3, D–F), even more so than early replicating regions, likely enabling them to start replication despite the decrease in CDC7 activity. Conversely, delays in RT were often found in large genes (>400 kb) (Fig. S3 C), which are known to be poor in replication origins and prone to replication stress, poor in putative G4 sequences and CpG islands, and rarely contain constitutive origins (Fig. S3, D–F).

These changes in RT are consistent with reduced origin activation, which favors a delay in those regions that are sparse in origins and rely on passive replication, while regions with a high density of origins have a higher chance of being activated even if CDC7 activity is partially compromised. Major changes in the RT were described in RIF1-deficient cells (Alver et al., 2017; Foti

et al., 2016). These are cell type dependent and are reinforced upon several rounds of replication, correlating with redistribution of chromatin marks and alterations in chromatin architecture (Klein et al., 2021). We observe analogies between the changes in RT upon CDC7 inhibition and DBF4 loss with the changes obtained in mouse embryonic stem cells after the loss of RIF1-PP1 interaction (Gnan et al., 2021). In both cases, the RT profiles show a higher degree of distinction between Early and Late replicating domains than in RIF1-KO, where the RT program is distributed solely toward mid-S-phase regions (Klein et al., 2021). This raises the possibility that protracted CDC7 inhibition may lead to epigenetic perturbation, a hypothesis that should be experimentally tested in future studies.

## DBF4 is required for checkpoint signaling and CDC7 activity at stalled forks

We reported that upon prolonged fork stalling, CDC7 inhibition suppresses checkpoint signaling and DNA double-strand break (DSB) induction (Rainey et al., 2020b). To understand if DBF4 or DRF1 mediates this role, we treated MCF10A EditR, DBF4-, and DRF1-deficient cells with hydroxyurea (HU) for 16 h; as a control, MCF10A EditR cells were also treated with XL413 (Fig. 4, A and B). XL413 treatment, as well as DBF4 loss, affected CHK1 phosphorylation at Ser345, a downstream marker of ATR activity, while in DRF1-deficient cells, this was not compromised (Fig. 4 A). Interestingly, ATR autophosphorylation was not obviously reduced (Fig. 4 B), suggesting that DBF4 with CDC7 mediates an intermediate step in signaling amplification, likely the well-characterized CDC7-dependent phosphorylation of CLASPIN, which facilitates CHK1 activation by ATR (Kim et al., 2008; Rainey et al., 2013; Yang et al., 2019).

Replication fork stalling by HU induces H2AX phosphorylation at Ser139 (γH2AX), which is further increased by replication fork collapse and DSB formation (Petermann et al., 2010). Intriguingly, γH2AX in HU appeared to be partially reduced in both DBF4- and DRF1-deficient cells and upon XL413 treatment (Fig. 4 B). With a more quantitative flow cytometry–based assay, we found that γH2AX induction was drastically reduced in DBF4-deficient cells and by XL413 treatment in parental cells, while the reduction in DRF1-deficient cells was much more limited (Fig. 4 C). At stalled forks, CDC7 promotes MRE11-dependent processing, and we previously identified a small pool of MRE11 displaying a CDC7-dependent, phosphatase-sensitive electrophoretic mobility shift in HU-treated cells (Rainey et al., 2020b). MCF10A EditR, either mock or treated with XL413, MCF10A DBF4-11, and DBF4-30, as well as MCF10A DRF1-7 cells were therefore treated with 4 mM HU for 24 h. MRE11 mobility shift in HU-treated cells was drastically reduced by XL413 and in DBF4-deficient cells, but again to a much lesser extent in DRF1-deficient cells (Fig. 5, A and B).

To assess the impact of DBF4 and DRF1 deficiency on CDC7 functions directly at replication forks, we extracted nascent DNA and analyzed associated proteins by DNA-mediated chromatin pull-down (Dm-ChP) (Kliszczak et al., 2011). In the first set of experiments, we found that CDC7 was recruited at replication forks in an XL413-independent manner, and it was

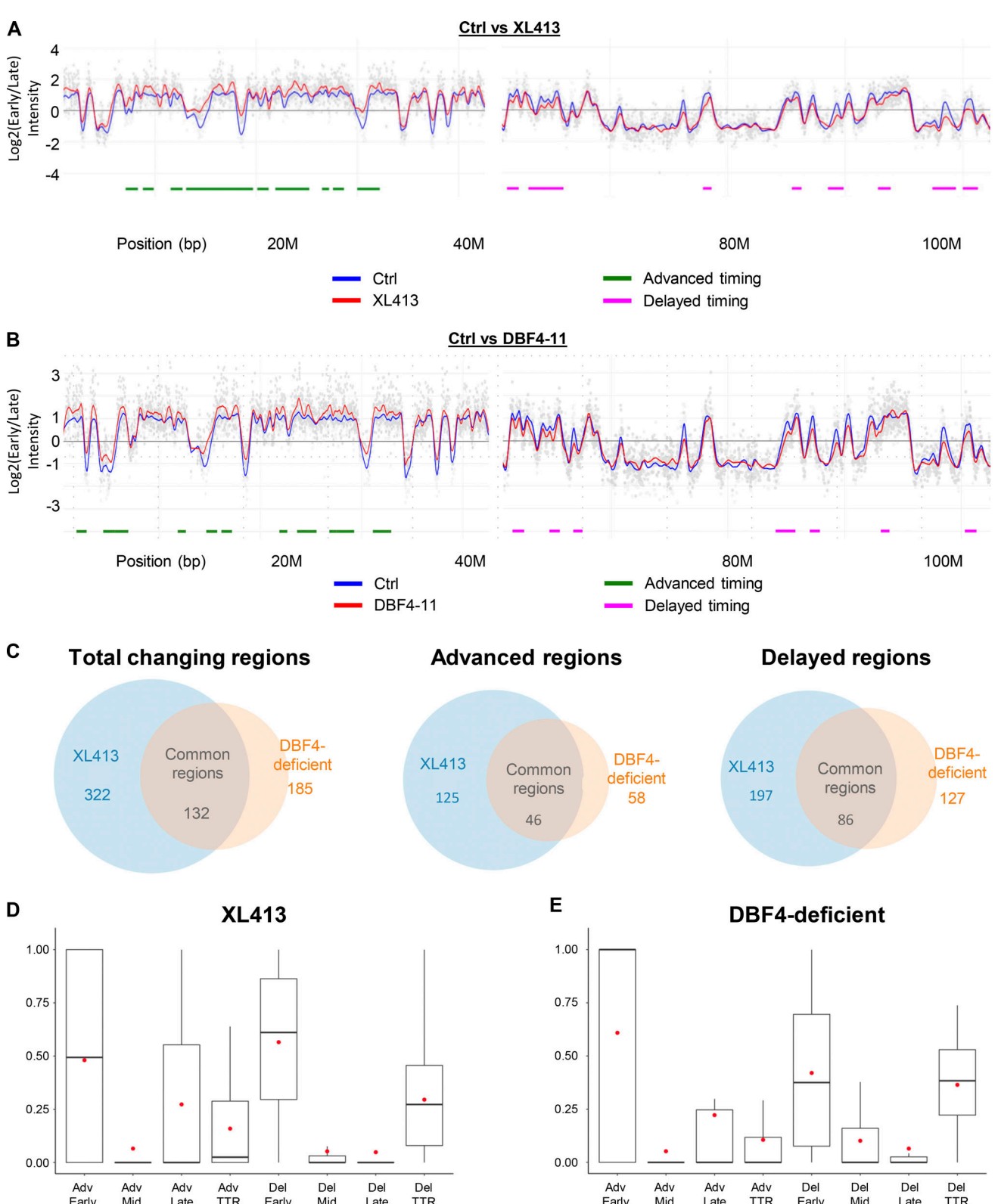

Figure 3. **Loss of DBF4 and CDC7 inhibition induces similar changes in global RT.** MCF10A EditR and MCF10A DBF4-deficient cells were treated with 10 µM XL413 or DMSO for 24 h before RT analysis. **(A)** Part of chromosome 1 RT profiles in untreated or XL413-treated MCF10A EditR cells. RT profiles display the log ratio between early and late replicated fractions along the chromosome. Positive log ratios correspond to early replicated regions whereas negative ones correspond to late replicated regions. The blue line represents MCF10A EditR cells treated with DMSO and the red one, cells treated with 10 µM XL413. Chromosome coordinates are indicated below the profile in megabases (M). Differences in RT are marked below the profile with advanced regions in green and delayed regions in magenta. Data is representative of two replicates of four independent experiments. **(B)** Part of chromosome 1 RT profiles for MCF10A EditR compared with MCF10A DBF4-deficient cells (DBF4-11). Analysis was performed and graphs were generated as described in A. **(C)** Summary of changes in RT

in MCF10A EditR treated with 10 µM XL413 (blue) and MCF10A DBF4-deficient cells (orange) displayed as a Venn diagram for total changing regions (left), advanced regions (middle), and delayed regions (right). Numbers represent the numbers of changed regions for indicated samples. Four independent experiments were performed for XL413-treated MCF10A cells and for DBF4-deficient cells. **(D)** Analysis of RT changes in MCF10A EditR treated with 10 µM XL413 for 24 h relative to the RT regions they originated from; either early, mid, and late replicating regions or TTR. Advanced regions (Adv) and delayed regions (Del) are displayed separately. The box plots show the dispersion of the data with a range from the 25th to 75th percentile, the sample median is represented by the line inside the box and the mean by a red dot. **(E)**. Analysis of RT changes in MCF10A DBF4-deficient cells treated with DMSO for 24 h as described in D.

present in both DBF4- and DRF1-deficient cells during unperturbed DNA replication (Fig. 5 C). We then treated cells with 4 mM HU over 24 h and observed that the mobility shift of MRE11 at forks was lost in both XL413-treated and DBF4-deficient cells correlating with the suppression of H2AX and

RPA2 phosphorylation (Fig. 5 D). Instead, in DRF1-deficient cells, MRE11 phospho-shift and H2AX and RPA2 phosphorylation were either not affected or only very partially compromised (Fig. 5 E), thus identifying DBF4 as the regulatory subunit involved in CDC7's function at forks.

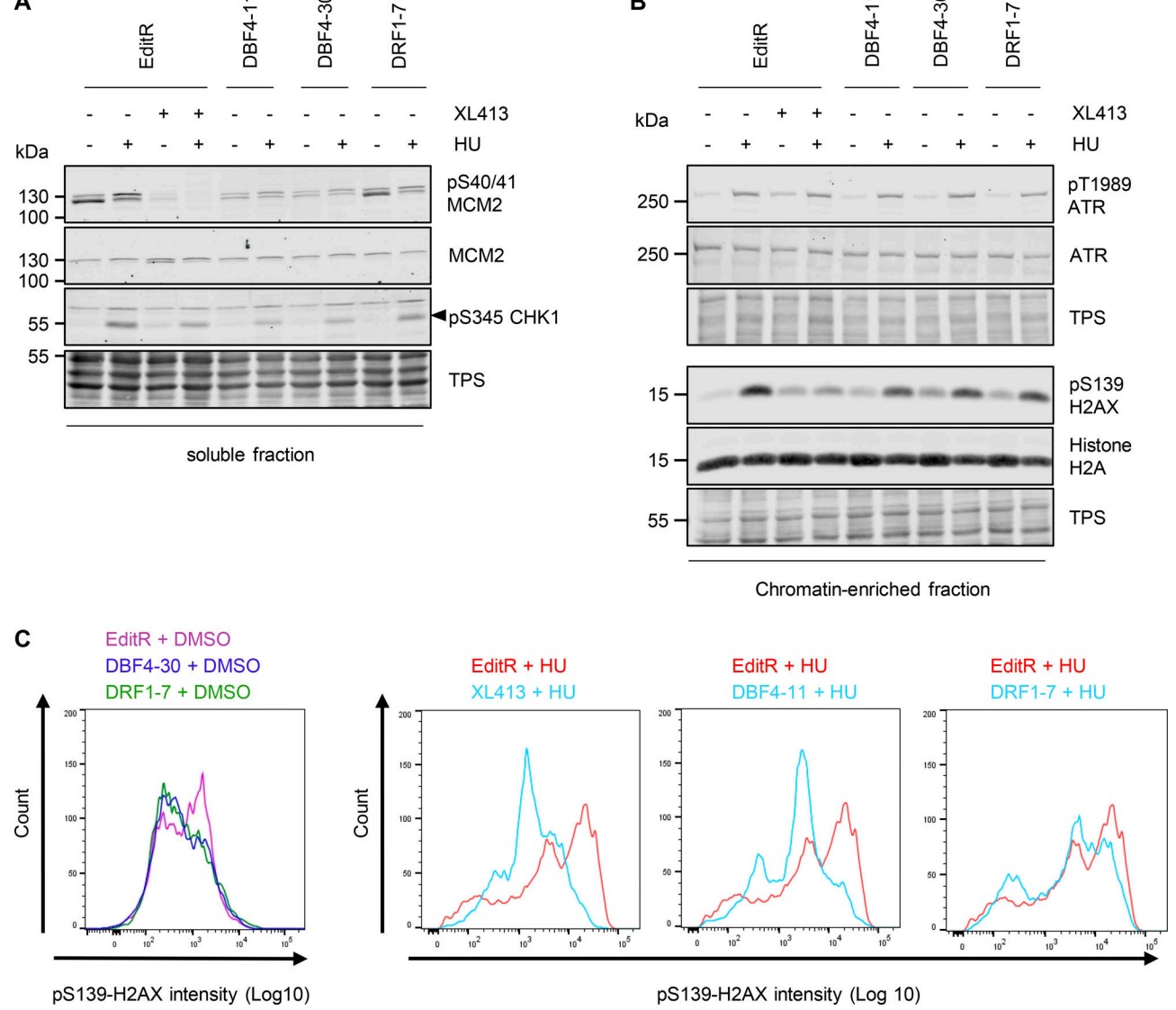

Figure 4. **DBF4 mediates CDC7 activity in the replication stress response. (A and B)** MCF10A EditR, MCF10A DBF4-11 and -30, and MCF10A DRF1-7 were pretreated with 10 µM XL413 or DMSO for 30 min before the addition of 4 mM HU for 16 h. Soluble (A) and chromatin-enriched (B) fractions were prepared and analyzed by immunoblotting with indicated antibodies. TPS was used as loading control. Triangle marks the pS345 CHK1 band. Data are representative of three independent experiments. **(C)** MCF10A EditR, MCF10A DBF4 -30, and MCF10A DRF1-7 were pretreated with 10 µM XL413 or DMSO for 30 min before the addition of 4 mM HU for 24 h. For flow cytometry analysis of pS139-H2AX intensity, cells were harvested, fixed, and stained with the indicated antibody. Representative images from one of three independent experiments are shown. MCF10A EditR treated with HU in the panels is the same sample in pairwise comparison with other samples for better visualization. Source data are available for this figure: SourceData F4.

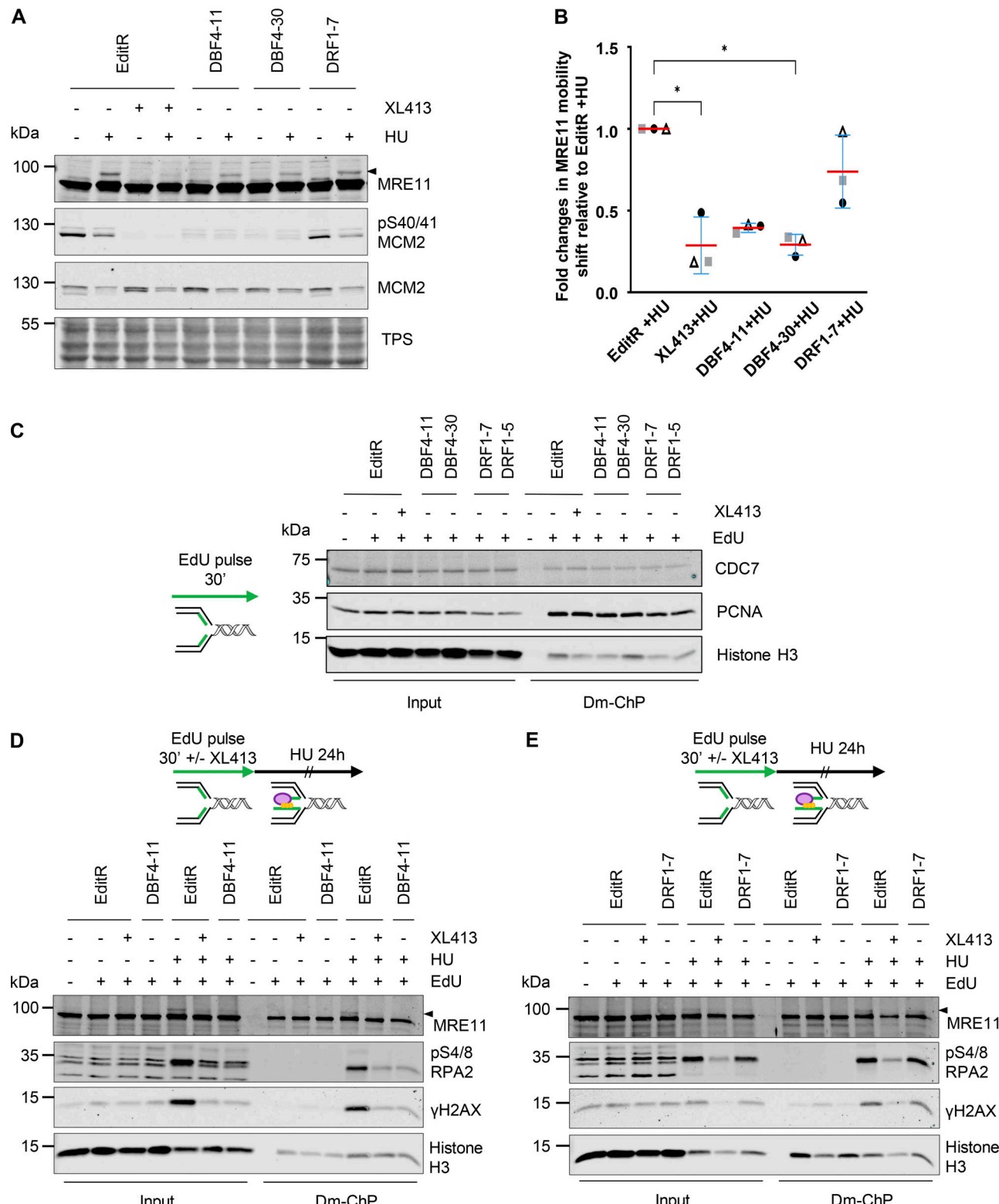

Figure 5.  **CDC7's activity at replication forks is solely mediated by DBF4. (A)** MCF10A EditR, MCF10A DBF4-11, and -30, and MCF10A DRF1-7 were pretreated with 10 µM XL413 or DMSO for 30 min before the addition of 4 mM HU for 24 h. Whole-cell extracts were prepared and analyzed by immunoblotting with indicated antibodies. TPS was used as loading control. Triangle marks MRE11 electrophoretic mobility shift. Data are representative of three independent experiments. **(B)** Quantification of MRE11 mobility shift. Data shows fold change in MRE11 mobility shift of HU-treated DBF4-11, DBF4-30, DRF1-7, and XL413-treated EditR samples relative to MCF10A EditR cotreated with DMSO and HU in three independent experiments. Data were normalized to TPS to account for differences in loading. Data are presented as mean fold change ± SD. Statistical analysis was performed using Kruskal–Wallis test, with Dunn's multiple comparison test (*<0.05). **(C)** MCF10A EditR, MCF10A DBF4-11, and MCF10A DRF1-7 were treated with 10 µM XL413 or DMSO for 24 h. Before

harvesting, cells were labeled with EdU for 30 min, then cells were fixed and proteins binding to EdU-labeled DNA were captured by Dm-ChP. A graphical representation of the experiment is shown on the left of the western blot analysis. Both input and captured material (Dm-ChP) were analyzed by western blot with indicated antibodies. Histone H3 was used as a loading control. Unlabeled cells (– EdU) were used as a negative control for the Dm-ChP samples. Data are representative of two independent experiments for EditR and clone DBF4-11 and once for the other clones. **(D)** MCF10A EditR and MCF10A DBF4-11 were pretreated with 10 µM XL413 or DMSO and 10 µM EdU for 30 min followed by a treatment with 4 mM HU for 24 h. Samples were prepared as described in C. Triangle marks MRE11 electrophoretic mobility shift. Data are representative of three independent experiments. **(E)** MCF10A EditR and MCF10A DRF1-7 were pretreated with 10 µM XL413 or DMSO and 10 µM EdU for 30 min followed by a treatment with 4 mM HU for 24 h. Samples were prepared as described in C. Triangle marks MRE11 electrophoretic mobility shift. Data are representative of two independent experiments. Source data are available for this figure: SourceData F5.

While the impairment of CHK1 signaling is likely attributable to CDC7/DBF4 phosphorylation of CLASPIN, the suppression of H2AX and RPA2 phosphorylation is instead consistent with weakened MRE11 activity in DBF4-deficient cells. Intriguingly, CDC7 recruitment at forks is unaffected by DBF4 or DRF1 deficiency; we speculate that CDC7 is either recruited by protein–protein interactions mediated by both DBF4 and DRF1, with only DBF4 being able to target the kinase to key substrates or is solely dependent on domains located on CDC7.

CDC7 functions have been primarily investigated without distinction between CDC7-DBF4 and CDC7-DRF1 complexes. In vitro, both proteins activate the kinase and can phosphorylate the MCM complex (Montagnoli et al., 2002). Similarly, the switch from DRF1 to DBF4 in *Xenopus laevis* over the course of its development (Collart et al., 2017; Takahashi and Walter, 2005) suggests that there is some redundancy between the two subunits in supporting the essential activity of CDC7 in DNA replication.

Recent work using targeted protein degradation and chemical biology approaches indicated that human cells can replicate their DNA and proliferate in the absence of CDC7 through compensation by CDK1 (Suski et al., 2022). In contrast, we and others showed that replication and proliferation are highly compromised by inhibition of either kinase, kinase-dead mutations, or by genetic ablation (Jones et al., 2021; Kim et al., 2002; Rainey et al., 2017). The idea that CDC7 is required for proliferation is reinforced by more than 1,000 CRISPR screens in different human cell lines defining *CDC7*, just like *CDK1*, as a core essential gene (Fig. 1 C and http://DepMap.org). In our view, the conflicting results about the essentiality of *CDC7* can be reconciled by acknowledging that human CDC7 is indeed necessary and that the total loss of kinase function is required to block DNA replication, which may have not been achieved in the Suski study. Incomplete CDC7 inhibition still allows genome duplication due to the large abundance and redundancy in replication origins, which means that activation of only a small fraction of these is sufficient for full genome replication, as similarly observed by depleting MCM proteins (Alver et al., 2014; Ge et al., 2007; Ibarra et al., 2008). Several proteins have now been shown to modulate the effectiveness of CDC7 inhibition including RIF1, PP1, ATR, and PTBP1 (Göder et al., 2023; Hiraga et al., 2017; Jones et al., 2021; Rainey et al., 2020a). Specifically, CDK1 phosphorylation disrupts RIF1/PP1 interaction and PP1's ability to counteract MCM phosphorylation by CDC7 (Moiseeva et al., 2019); thus CDK1 activity can help in dealing with decreased levels of CDC7 kinase, as we have previously shown (Rainey et al., 2017).

We found that it is DBF4, more than DRF1, that mediates CDC7 activity in DNA replication and replication stress response. This preference for DBF4 may be quantitative, i.e., higher DBF4 than DRF1 protein levels in cells or due to structural features affecting the strength of key protein–protein interactions, thus driving CDC7 to its targets with different affinities.

CDC7 inhibitors, while very effective preclinically as anticancer drugs, have had limited success in clinical trials so far. The direct targeting of DBF4 with technologies such as "molecular glues" and genome editing approaches may provide an alternative with a better therapeutic window by targeting cancer types specifically dependent on DBF4 expression and avoiding the blockade of DNA replication in normal cells.

## Materials and methods

### Cell culture

Cell culture was performed in a Class II Bio-safety cabinet, and all cell lines were maintained at 37°C in a humidified atmosphere containing 5% $CO_2$. Cell counts and viability were determined using Trypan blue exclusion and a countess (Invitrogen) or LUNA II cell counter (Logos biosystems). MCF10A cell line (human, female origin) was purchased from ATCC (Cat# CRL-10317; RRID:CVCL_0598) and authenticated via whole-genome sequencing. MCF10A EditR cells stably express *Streptococcus pyogenes* Cas9 nuclease (Rainey et al., 2017). MCF10A DBF4-deficient clones 11 and 30, as well as DRF1-deficient clones 5 and 7, were derived via CRISPR/Cas9 genome editing from MCF10A EditR cells and monoclonal expansion. The mutation of *DBF4* or *DRF1* was verified via targeted PCR and Sanger sequencing. MCF10A cells and derivatives were cultured using DMEM supplemented with 5% (vol/vol) horse serum, 25 ng/ml cholera toxin, 10 µg/ml insulin, 20 ng/ml epidermal growth factor (Peprotech), 500 ng/ml hydrocortisone, 50 U/ml penicillin, and 50 µg/ml streptomycin. Lenti-X 293T cell lines (human, female origin) were purchased from TaKaRa and used without further authentication. Cell culture was performed using DMEM + GlutMAX-I (Thermo Fisher Scientific) supplemented with 10% (vol/vol) fetal bovine serum, 50 U/ml penicillin, and 50 µg/ml streptomycin. Unless otherwise specified, all culture media and reagents were obtained from Merck.

### Drugs treatments

If not otherwise indicated, XL413 (synthesized in-house) was used at a concentration of 10 µM. HU was used at 4 mM and nocodazole at 0.2 µg/ml (all acquired from Merck).

## DNA transfections

Cells were transfected with plasmid DNA (0.75 µg) using a 1:3 (wt/vol) ratio of DNA to polyethyleneimine "MAX" MW 40,000 (1 mg/ml; Polysciences). These were first individually diluted in 100 µl of 150 mM NaCl, mixed, and incubated at room temperature for 20 min before being added to the cells.

## Generation of DBF4- and DRF1-deficient monoclonal cell lines

Oligonucleotides coding for sgRNAs targeting *DBF4* (5′-TGGGTC GAATTTCTCCTGTA-3′) or *DRF1* (5′-CGTTCCTCAAAATCGAAG AT-3′) were cloned into Bbs1 restriction site of the pX330-Hygromycin plasmid. MCF10A EditR cells were transiently transfected with pX330-Hygromycin vectors carrying DBF4-sgRNA and DRF1-sgRNA. Cells were cultured for 24 h before media change and selection with hygromycin B (50 µg/ml) for a further 72 h. The surviving cells were trypsinized, subjected to limited dilution, and incubated in 96-well plates for 10–14 days. Cells from wells containing only single colonies were expanded and genotyped to assess the mutational status of *DBF4* or *DRF1*. For screening, cells were washed with PBS and lysed in 50 µl of lysis buffer (10 mM Tris-HCl, pH 7.5, 10 mM EDTA, 10 mM NaCl, 0.5% [wt/vol] N-Lauryl sarcosine, 10 µg/ml Proteinase K, and 20 µg/ml glycogen) for 2 h at 60°C. Genomic DNA was precipitated by adding 3× volumes of 150 mM NaCl in 96% (vol/vol) EtOH before mixing and incubation at room temperature for 30 min. DNA was pelleted by centrifugation (15 min, 16,000 × *g*), washed with 70% (vol/vol) EtOH, and repelleted prior to air drying and resuspension in 100 µl TE buffer (10 mM Tris-HCl, pH 7.5, 1 mM EDTA). 5 µl genomic DNA was used in PCR reactions using Taq DNA Polymerase and screening primers (DBF4-fwd: 5′-ACTTTGTTCTCTTCTAGCGAGTTG-3′, DBF4-rev: 5′-CGCCATCCCTAAATACAAGGGT-3′; DRF1-fwd: 5′-GGCTCT TAGGCTTTGGCAGA-3′, DRF1-rev: 5′-GTCAGCATACACCCAAGG GG-3′). PCR products were purified with MACHEREY-NAGEL NucleoSpin Gel and PCR clean-up kit and sequenced using DBF4-fwd or DRF1-fwd primer by Eurofins genomics. Genome editing was accessed by manual analysis of the Sanger sequencing reads.

## Protein samples preparation

For whole-cell extracts, cells were harvested, washed with PBS, and resuspended in 1 volume of 20% (vol/vol) trichloroacetic acid (TCA). Samples were mixed and two volumes of 5% (vol/vol) TCA were added before centrifugation at 3,000 rpm for 10 min. Pellets were resuspended in appropriate volumes of 2× Laemmli buffer, pH of the extracts was neutralized using 1 M Tris base, and samples were denatured for 5 min at 95°C. Proteins were separated by SDS-PAGE.

For chromatin fractionation, MCF10A cells were seeded at 150,000 per well in 6-well plates, and following treatment with indicated reagents, cells were harvested, washed once with PBS, and lysed in CSK buffer (10 mM PIPES, pH 6.8; 300 mM sucrose; 100 mM NaCl; 1.5 mM MgCl₂; 0.5% [vol/vol] Triton X-100; 1 mM ATP; 1 mM DTT; 1 mM sodium orthovanadate; 2 mM N-ethylmaleimide; Phosphatase Inhibitor Cocktail I and Protease Inhibitor Cocktail III [Thermo Fisher Scientific]) for 10 min on ice. Samples were centrifuged at 1,000 × *g* for 5 min at 4°C and the supernatant was transferred to a new reaction tube

(soluble fraction). The pellet was washed using CSK buffer (2× volume of soluble fraction), centrifuged at 1,000 × *g* for 5 min at 4°C, and the supernatant was discarded. Pellets were resuspended in CSK buffer containing benzonase (125 U/ml) and incubated for 30 min on ice. Samples were denatured for 5 min at 95°C in 1× Laemmli buffer (chromatin-enriched fraction). Protein concentration of the soluble fraction was determined by Bradford assay, and the required amount of protein was denatured for 5 min at 95°C in 1× Laemmli buffer.

## Immunoblotting

Proteins were transferred onto 0.2-µm-pore-size nitrocellulose membranes using a wet blot transfer system (Biorad). Proteins on membranes were stained with fast green (0.0001% [wt/vol] fast green in 0.1% [vol/vol] acetic acid) for 5 min as a total protein stain (TPS) and were analyzed on the Odyssey infrared imaging system at 680 nm (LI-COR Biosciences). Membranes were destained with 0.1 M NaOH in 30% (vol/vol) methanol for 10 min and washed three times in double-distilled water (ddH₂O) for 5 min at room temperature. Membranes were blocked in 3% (wt/vol) skim milk (Sigma-Aldrich) in TBS-T (20 mM Tris-HCl, pH 7.5, 150 mM NaCl, 0.05% [vol/vol] Tween-20) for 1 h at room temperature. Membranes were incubated in primary antibody diluted in blocking buffer overnight at 4°C followed by three washes in TBS-T for 10 min each. Secondary antibodies were diluted in blocking buffer and membranes were incubated for 1 h at room temperature (protected from light) followed by three washes in TBS-T for 10 min at room temperature. Signals were acquired using the Odyssey infrared imaging system and analyzed using Image Studio 2.0.38 and Empiria software 1.3.0.83 (LI-COR).

Primary antibodies were diluted in 3% skim milk/TBS-T: CHK1 (sc8408. RRID:AB_627257 1:1,000; Santa Cruz Biotech), CDC7 (DCS-342, RRID:AB_591045; 1:1,000; MBL), DBF4 (JDi74 from John Diffely's lab: raised against a C-terminus fragment of DBF4; 1:1,000), MCM2 (in-house: 1:3,000; Natoni et al., 2013), or 1% BSA/TBS-T: pT1989 ATR (GTX128145, RRID:AB_2687562; 1:1,000; GeneTex), pS345 CHK1 (2348, RRID:AB_331212, 1:1,000; CST), pS139 H2AX (9718, RRID:AB_2118009, 1:1,000; CST), pS4/8 RPA32 (A300-254A, RRID:AB_210547, 1:1,000; Bethyl Laboratories), GAPDH (SC-25778, RRID:AB_10167668; 1:3,000; Santa Cruz Biotech), and pS40/S41 MCM2 (in-house; 1:3,000; Montagnoli et al., 2006). IRDye secondary antibodies (LI-COR): 800CW goat anti-rabbit (926-32211, RRID:AB_621843; 1:10,000; LI-COR Biosciences) and 800CW goat anti-mouse (926-32210, RRID: AB_621842; 1:10,000; LI-COR Biosciences) were diluted in the same buffer as the primary antibody.

## Immunoprecipitations (IP)

For co-IP experiments, cell lysates were prepared in CSK buffer (300 mM NaCl, 10 mM PIPES, pH 6.8, 300 mM sucrose, 1.5 mM MgCl₂, 1 mM dithiothreitol, 0.5% Triton X-100) supplemented with protease and phosphatase inhibitors (Thermo Fisher Scientific). Mouse IgG (Sigma-Aldrich) or anti-DRF1 mAb 5G4 (Montagnoli et al., 2002) were prebound to 30 µl of protein A/G beads (Santa Cruz Biotechnology), and 1 mg of precleared lysate was immunoprecipitated for 2 h with rotation at 4°C. Following

5× washes with CSK buffer, proteins were recovered in 1× Laemmli buffer, heated at 95°C for 3 min, and analyzed alongside 20 μg of input material by SDS-PAGE and western blotting.

### Dm-ChP

For analysis of proteins that were associated with nascent DNA, cells were plated at $9 \times 10^6$ cells in 150-mm plates. Following treatment, cells were labeled with 10 μM EdU for 30 min, processed, and then analyzed using the Dm-ChP technique (Kliszczak et al., 2011).

On the day before harvesting, streptavidin agarose beads (10302384; Thermo Fisher Scientific) were prepared (100 μl per sample, 50% slurry) by washing the required amount of beads three times with 1 ml of Wash buffer (10 mM Tris-HCl, pH 8.0, 140 mM NaCl, 0.5 mM DTT) and centrifuged for 2 min at 1,200 rpm and 4°C. Then, beads were blocked overnight at 4°C (under constant rotation) with 1 ml of blocking buffer (0.5 mg/ml BSA and 0.4 mg/ml presheared salmon sperm DNA in radioimmunoprecipitation assay [RIPA] buffer [10 mM Tris-HCl pH 8.0, 140 mM NaCl, 0.1% (wt/vol) sodium deoxycholate, 0.1% (wt/vol) SDS, 1% (vol/vol) Triton X-100 with Phosphatase Inhibitor Cocktail I, and Protease Inhibitor Cocktail III]) to minimize nonspecific binding. Beads were centrifuged for 2 min at 1,200 rpm and 4°C, the supernatant was discarded, and the beads were transferred to a new tube. The beads were washed twice with 1 ml Wash buffer and centrifuged for 2 min at 1,200 rpm and 4°C. For the last wash, beads were resuspended in 500 μl of Wash buffer, transferred to a new tube, and stored at 4°C until cell lysates were prepared. Before use, beads were washed one last time (2 min at 1,200 rpm and 4°C) and resuspended in an appropriate volume RIPA buffer to achieve 50% slurry.

Following labeling with EdU, cells were washed twice with PBS before fixing the cells for 10 min at room temperature using serum-free media containing 1% (wt/vol) PFA. To quench the PFA, 0.125 M Glycine was added to the plates and cells were incubated for 10 min at room temperature on a shaker. The media was discarded and plates were washed three times with 10 ml of ice-cold PBS. In the following, cells were scraped off the plates using a cell scraper and transferred into a 15-ml falcon tube. The samples were centrifuged (1,200 rpm, 5 min, 4°C), the supernatant was removed, and cells were permeabilized with the addition of 1 ml 0.1% (vol/vol) Triton-X100 in PBS with Phosphatase Inhibitor Cocktail I and Protease Inhibitor Cocktail III (Thermo Fisher Scientific) for 10 min on ice. Triton/PBS solution was removed (1,200 rpm, 5 min, 4°C) and washed once with ice-cold PBS before performing a click reaction by gently resuspending the samples in 1 ml of Click reaction mix (10 mM Sodium-L-ascorbate, 0.1 mM Biotin-TEG azide, 2 mM CuSO₄, in PBS) and incubating them in the dark for 30 min at room temperature. After the incubation, 10 ml of PBS-T (0.5% [vol/vol] Tween-20; 1% BSA [wt/vol] in PBS) was added and samples were incubated for an additional 10 min to remove excess copper and azide. Samples were washed in PBS (1,200 rpm, 5 min, 4°C) and cell pellets were resuspended in 1.2 ml of Cytoplasmic Lysis buffer (50 mM HEPES, pH 7.8, 150 mM NaCl, 1.5 mM MgCl₂, 0.5% [vol/vol] NP-40, 0.25% [vol/vol] Triton X-100, 10% [vol/vol] glycerol) and transferred to 1.5-ml Eppendorf tubes.

Samples were rotated for 10 min at room temperature, before being centrifuged at 1,300 rpm for 5 min. The supernatant was collected as the soluble fraction. The pellets were washed with 1.2 ml Wash buffer and rotated for 10 min at room temperature before being centrifuged at 1,300 rpm for 5 min at room temperature. The pellets were resuspended in 1.2 ml RIPA buffer and incubated under constant rotation for 5 min at room temperature. In the next step, samples were sonicated using a digital sonifier (Branson) according to the following conditions: 40% amplitude, 1 s pulse on, 10 s pulse off, and 10 cycles. This sonication was repeated six times for each sample to fragment the labeled DNA. After sonication, samples were centrifuged at $12,000 \times g$ for 10 min at 4°C and the supernatant was transferred into a new tube. The protein concentration of the samples was quantified using a BCA assay kit (Thermo Fisher Scientific). Equivalent amounts of protein were removed for each sample and RIPA buffer was added to a final volume of 1 ml. Additionally, 15–20 μg of protein was removed as input sample for later use. 100 μl of preblocked streptavidin agarose beads were added to each sample before incubation overnight at 4°C and constant rotation (12 rpm). After the incubation, the IP samples were centrifuged at 1,200 rpm for 2 min and supernatant was removed as flowthrough. The samples (beads) were washed six times with 1 ml of wash buffer and then resuspended in 70 μl of RIPA buffer with 1× Laemmli buffer. Samples were incubated at 95°C for 5 min and the IP eluate was carefully removed using a Hamilton syringe. Samples were analyzed via SDS-PAGE and western blot analysis.

### RT analysis

20,000,000 exponentially growing cells were incubated and protected from light with 50 μM BrdU (#142567; Abcam) at 37°C for 90 min. Cells were then fixed in 75% cold EtOH and stored at −20°C. BrdU-labeled cells were incubated with 80 μg/ml propidium iodide (P3566; Invitrogen) and 0.4 mg/ml RNaseA (10109169001; Roche) for 1 h at room temperature. 150,000 cells were sorted in early (S1) and late (S2) S-phase fractions using a fluorescence-activated cell sorting system (FACS Aria Fusion; BD) in lysis buffer (50 mM Tris, pH 8, 10 mM EDTA, 0.5% SDS, 300 mM NaCl). DNA from each fraction was extracted using Proteinase K treatment (200 μg/ml, EO0491; Thermo Fisher Scientific) followed by phenol–chloroform extraction and sonicated to a size of 500–1,000 bp (Hadjadj et al., 2016). IP was performed using the IP star robot at 4°C (indirect 200 μl method; SX-8G IP-Star Compact Automated System, Diagenode) with an anti-BrdU antibody (10 μg, purified mouse anti-BrdU, #347580; BD Biosciences). Denatured DNA was incubated for 5 h with anti-BrdU antibodies in IP buffer (10 mM Tris, pH 8, 1 mM EDTA, 150 mM NaCl, 0.5% Triton X-100, 7 mM NaOH) followed by an incubation of 5 h with Dynabeads Protein G (10004D; Invitrogen). Beads were then washed with Wash buffer (20 mM Tris, pH 8.0, 2 mM EDTA, 250 mM NaCl, and 1% Triton X-100). Reversion was performed at 37°C for 2 h with a solution containing 1% SDS and 0.5 mg Proteinase K followed, after the bead's removal, by incubation at 65°C for 6 h in the same solution.

Immunoprecipitated BrdU-labeled DNA fragments were extracted with phenol–chloroform and precipitated with cold ethanol. Control qPCRs were performed using oligonucleotides specific to mitochondrial DNA, early (BMP1 gene), or late (DPPA2 gene) replicating regions (Hadjadj et al., 2016). Whole-genome amplification was performed using the SeqPlex Enhanced DNA Amplification kit as described by the manufacturer (SEQXE; Sigma-Aldrich). Amplified DNA was purified using a PCR purification product kit as described by the manufacturer (740609.50; Macherey-Nagel). DNA amount was measured using a Nanodrop. Quantitative PCRs using the oligonucleotides described above were performed to check whether the ratio between early and late replication regions was still maintained after amplification. Early and late nascent DNA fractions were labeled with Cy3-ULS and Cy5-ULS, respectively, using the ULS arrayCGH labeling Kit (EA-005; Kreatech). The same amounts of early- and late-labeled DNA were loaded on human DNA microarrays (SurePrint G3 Human CGH arrays, G4449A; Agilent Technologies). Hybridization was performed at 65°C. The following day, microarrays were scanned using an Agilent C-scanner with Feature Extraction 9.1 software (Agilent technologies). The START-R suite was used to analyze the data (Hadjadj et al., 2020). Differential analysis of two experiments, each composed of two technical replicates, were performed with START-R analyzer and visualized with START-R viewer.

### Genomic studies of advanced and delayed RT domains
For each experiment, START-R Analyzer generated segmentation bed files corresponding to early, mid, late, TTR, advanced, and delayed replicating domains. We also generated a random sample on the Galaxy website from the initial file containing the genomic coordinates of all advanced and delayed replicating domains detected after CDC7 inhibition and DBF4 loss using successive rounds of Galaxy bedtools ShuffleBed randomly redistribute intervals in a genome (Galaxy Version 2.30.0) to obtain a random sample of 50,700 genomic regions.

The coverage of the different replicating domains and the random sample with large genes (>400 kb), constitutive origins, CpG islands, and putative G4 were done with the coverage of a set of intervals on the second set of intervals software (Galaxy Version 1.0.0). Boxplots illustrating differences in these coverages were generated. The constitutive origins and the putative G4 files (hg19 genome assembly) were taken from Picard et al. (2014) and converted with LiftOver to hg18 genome assembly. The position of genes came from the UCSC table browser RefSeq Genes database without duplicates and with hg18 genome assembly. Large gene regions (>400 kb) were extracted from the aforementioned gene database. The CpG islands file came from the UCSC browser (hg19 assembly) and was converted with LiftOver to hg18 genome assembly. Data are deposited in GEO with accession number GSE248981.

### Fluorescence microscopy
MCF10A cells were seeded at a density of 130,000 cells per well in a 6-well plate on poly-L-lysine–coated coverslip. Following drug treatment, coverslips were washed once with PBS, and cells were fixed in PBS containing 4% (wt/vol) PFA for 10 min

at room temperature. Following the fixation step, samples were washed three times with PBS to remove residual PFA. Cells were permeabilized with PBS-TX (0.1% [vol/vol] Triton X-100 in PBS) for 20 min at room temperature followed by incubation for 30 min in blocking buffer (10% [vol/vol] FBS and 0.5% [wt/vol] BSA in PBS-TX) at room temperature. Cells were incubated for 1 h at 37°C with mouse anti-β-Tubulin (#05-661; 1:1,500; Millipore) primary antibody diluted in blocking buffer. Following three washes with PBS-TX, coverslips were incubated for 1 h at 37°C with goat anti-mouse AlexaFluor 488 (A11001; 1:500; Thermo Fisher Scientific) secondary antibody diluted in blocking buffer, which additionally contained DAPI (1:600 dilution in PBS of 0.5 mg/ml stock) to stain nuclei. Cover slips were washed three times in PBS-TX, once in PBS, and dipped in ddH$_2$O before being mounted onto slides using SlowFade Gold Antifade Reagent (Thermo Fisher Scientific).

Microscopy for micronuclei detection was performed using IX71 Olympus microscope with Olympus, UplanSApo, 60x/1.35 Oil, ∞/0.17/FN26.5 and Olympus, UApoN340, 40x/1.35 Oil, ∞/0.17/FN22 objectives. Images were captured at room temperature with a QImaging, Retiga R1 CCD Camera, and Volocity software (Quorum Technologies Inc.). The number of micronuclei per cell was manually counted using DAPI staining. β-Tubulin staining was used as cytoplasmic staining to assist in assigning micronuclei to specific cells. A minimum of 275 cells per sample of four independent experiments were analyzed. After quantification and analysis, the brightness of representative images was adjusted to the same extent for all samples in an experiment to aid visualization in figures.

### DNA combing
Cells were seeded at a density of 300,000 cells per well in a 6-well plate and allowed to recover overnight. To assess the replication fork speed in MCF10A cells in the presence of CDC7i or upon mutation of DBF4, cells were treated with DMSO or 10 μM XL413 for 24 h prior to labeling with 25 μM IdU for 30 min. Following the incubation, the media was exchanged and cells were washed three times with prewarmed culture media followed by treatment with DMSO or 10 μM XL413 and labeling with 250 μM CldU for 30 min. Immediately following the CldU pulse, 1 mM thymidine was added to block further CldU incorporation. Cells were harvested and washed with ice-cold PBS (5 min, 2,000 rpm, 4°C). The cell pellets were resuspended in 110 μl ice-cold PBS, counted using Trypan blue exclusion, and the cell number was adjusted to 1.5 × 10$^6$ cells/ml. A 1.5% (wt/vol) low-melting point agarose solution (in PBS) was prepared and stored at 45°C until required. 100 μl of the prepared cell suspension (=1.5 × 10$^5$ cells) was incubated at 45°C for 1 min before gently mixing it with 100 μl of the agarose solution and immediately transferring the mixture into two plug molds. The agarose plugs were allowed to set for 5 min at room temperature and then transferred to 4°C for 30 min to solidify. For cell lysis, plugs were incubated in 0.3 ml Proteinase K buffer (10 mM Tris-HCl, pH 7.5, 50 mM EDTA, 1% [wt/vol] Sarkosyl, 2 mg/ml Proteinase K) overnight at 50°C. The next day, plugs were transferred into 10 ml TE50 buffer (10 mM Tris-HCl, pH 7.5, 50 mM EDTA) and stored at 4°C until further use. To melt the agarose plugs, each

plug was washed two times in 10 ml TE-Wash buffer (10 mM Tris-HCl, pH 7.5, 1 mM EDTA) for 1 h with rotation at room temperature, before being transferred into 2 ml MES buffer (0.5 M MES hydrate, pH 5.5) for 1 h at 68°C. The resulting solution was cooled down to 42°C before adding 2 U of β-agarase (NEB), gently mixing the solution by end-over-tube inversion, and incubating it overnight at 42°C. Before combing, the samples were incubated at 68°C for 10 min, cooled down to room temperature, and transferred into Teflon reservoirs. Silanized coverslips were inserted into the combing apparatus and incubated in the DNA solution for 10 min before being automatically withdrawn by the DNA combing apparatus at a speed of 300 μm/s. Following the combing, slides were mounted onto glass slides and DNA was crosslinked to the slides by incubating them for 4 h at 60°C followed by storage at –20°C overnight. Slides were allowed to reach room temperature and combed DNA was denatured for 15 min in 0.05 M NaOH. Slides were washed three times in ice-cold PBS for 1 min to neutralize NaOH and then dehydrated by incubating the slides in 70, 90, and 100% EtOH in succession for 3 min each. After allowing the slides to dry at room temperature (in the dark), samples were blocked for 15 min in blocking buffer (1% [wt/vol] BSA/PBS). Next, slides were taken through a series of 30 min primary and then secondary antibody incubations in blocking buffer at room temperature, with three PBS washes in between each antibody incubation. The antibodies were used in the following order and concentrations: BrdU (BU1/75) rat monoclonal antibody (MA1-82088; 1:100; Thermo Fisher Scientific) and Chicken anti-rat Alexa Fluor 488 (A21470; 1:300; Thermo Fisher Scientific) to detect incorporated CldU; anti-BrdU (B44) IgG1 mouse monoclonal (347580; 1:100; BD Biosciences) and Goat anti-mouse IgG1 Alexa Fluor 546 (A21123; 1:300; Thermo Fisher Scientific) to detect incorporated IdU; anti-ssDNA poly dT (mab3034, 1:100; Millipore) and Goat anti-mouse IgG2a Alexa Fluor 647 (A21241; 1:300; Thermo Fisher Scientific) for general DNA fiber staining. After the final antibody, coverslips were washed two times in PBS, once in ddH$_2$O, and allowed to dry completely (protected from light). Coverslips were mounted onto the silanized coverslips using SlowFade Gold Antifade Reagent (Thermo Fisher Scientific). Images were captured with an IX71-Olympus microscope with an Olympus, UplanSApo, 60x/1.35 Oil, ∞/0.17/ FN26.5 objective. Images were captured at room temperature with a QImaging, Retiga R1 CCD Camera, and Volocity software (Quorum Technologies Inc.). The analysis was performed manually in ImageJ Fiji software.

**Flow cytometry**
Cell cycle distribution and rate of DNA synthesis were analyzed by plating MCF10A cells at 150,000 per well in 6-well plates. Following treatment with indicated inhibitors or DMSO, nascent DNA was labeled by incubating cells with 10 μM EdU for 30 min at 37°C prior to harvesting. Cells were washed with PBS, which if not specified otherwise, involves centrifugation at 400 × $g$ for 5 min at 4°C and the removal of the supernatant from the cell pellet. Samples were fixed by resuspension in 0.3 ml PBS and dropwise addition of 0.7 ml 100% ethanol while vortexing followed by incubation for at least 1 h at –20°C. After a PBS wash,

cells were washed with 1% (wt/vol) BSA/PBS and incorporated EdU was labeled by incubating cells in click reaction buffer (PBS containing 10 mM Sodium-L-ascorbate, 2 mM Copper-II-Sulfate, and 10 μM 6-Carboxyfluorescine-TEG-azide) for 30 min at room temperature protected from light. Cells were then incubated in PBS containing 1% (wt/vol) BSA and 0.5% (vol/vol) Tween-20 for 5 min at room temperature followed by an additional wash with PBS. DNA was stained with 1 μg/ml DAPI in 1% (wt/vol) BSA/PBS and to reduce RNA interference, 0.5 μg/ml RNase A was added to each sample until measurement. Fluorescence intensity data for DAPI (405_450_50 nm) and 6-carboxyfluorescine (488_530_30 nm) were acquired for 10,000 single cells on a BD FACS Canto II and analyzed using FlowJo 10.0.7 software. To perform an analysis of fluorescence intensity proportional to EdU incorporation for individual cells, gates were applied on DAPI-EdU biparametric dot plots to select for EdU-positive (late) S-phase cells using FlowJo. Fluorescence intensity (488_530_30-H) values per cell were exported and plotted as scatter dot plots using GraphPad Prism 10.0.2.

Detection of pS139 H2AX via flow cytometry was performed by plating 400,000 MCF10A cells on 6-cm plates and treating them as described for the respective experiments. Following the treatment, MCF10A cells were detached using trypsin/EDTA and washed once in ice-cold PBS (450 g, 5 min, 4°C). The washing steps were performed with the same centrifuge settings if not described otherwise. Next, cells were incubated in 300 μl 0.2% (vol/vol) Triton X-100/PBS for 10 min on ice followed by a washing step with PBS to remove excess Triton X-100 (450 g, 5 min, 4°C). Cell pellets were resuspended in 0.5 ml 1% (wt/vol) PFA/PBS and incubated for 10 min at room temperature. Cells were first washed in blocking buffer (1% [wt/vol] BSA/PBS) before being incubated for 30 min at 4°C in blocking buffer. The blocking buffer was removed (450 g, 5 min, 4°C) and cells were permeabilized by incubation in 0.05% (wt/vol) saponin buffer for 10 min on ice. In the following, samples were sequentially incubated with a primary antibody against pS139 histone H2AX (1:500, #9718; CST) for 2 h, followed by washing in saponin buffer, and then incubation with a secondary antibody (1:250, Donkey anti-rabbit Alexa Fluor 488; A21206; Thermo Fisher Scientific) for 30 min in the dark. Following two washing steps, cells were resuspended in 0.5 ml blocking buffer containing 0.1 mg/ml RNase and 1 μg/ml DAPI and incubated for 30 min in the dark at room temperature before measurement at the flow cytometer. Measurements and analysis were performed with the tools and settings described for EdU/DAPI staining.

**siRNA transfections**
MCF10A cells were seeded at 60,000 cells per well in a 6-well plate and transfected 24 h after plating. siRNA transfections were performed using JetPRIME transfection reagent (Polyplus Transfection). For each individual well, a pool of four siRNAs (50 or 100 nM final concentration, as indicated) was prepared by mixing with JetPrime buffer (200 μl) and JetPrime reagent (4 μl), and the mixture was incubated for 15 min at room temperature prior to dropwise addition to the cells. Following a 5 h incubation at 37°C, 5% CO$_2$ the culture media was exchanged for fresh, incubator-equilibrated media, and cells were returned to

the incubator. In this study, treatment of siRNA-transfected cells was performed 48 h after transfection. Protein depletion was confirmed by qPCR. The following siRNA sequences were used: siCtrl: 5′-AGUACUGCUUACGAUACGG-3′; siDBF4_1: 5′-GAACACACAUUAAGUGAAA-3′; siDBF4_2: 5′-GAGCAGAAU UUCCUGUAUA-3′; siDBF4_3: 5′-GCACAAACCUUGGGUCGAA-3′; siDBF4_4: 5′-CCAAACAGAUGGCGAUAAG-3′; siDRF1_1: 5′-GGAAACAUCGGCCAUGGUU-3′; siDRF1_2: 5′-GGAAACCCG UUGACUCGGU-3′; siDRF1_3: 5′-AAACAUCGGCCAUGGUUGA-3′; siDRF1_4: 5′-GAGCGAACCGGGAAAGGGA-3′.

### Real-time qPCR

MCF10A EditR cells were transfected with siRNA as described in "siRNA transfections" 72 h before total RNA was extracted using the NucleoSpin RNA columns (Machery Nagel) according to manufacturer instructions. The concentration of the total RNA was determined using a NanoDrop. To generate cDNA, 1 µg of RNA was used with random hexamer primers in the SuperScript First Strand Synthesis System (Thermo Fisher Scientific). The resulting cDNA was diluted 1:10 and used for real-time qPCR using the following TaqMan expression assays: Hs00272696_m1 (DBF4), Hs010691951_m1 (DRF1), Hs99999901_s1 (18S). The relative mRNA levels were normalized to the 18S rRNA expression and calculated using the comparative Ct method.

### Statistical analyses

We used GraphPad Prism 10.0.2 to perform most statistical analyses. The specific statistical tests used are indicated in the figure legends. For most experiments, we performed one-way ANOVA with Tukey's multiple comparison test for statistical evaluation. We assumed a Gaussian distribution for one-way ANOVA, but this was not formally tested. For data normalized to the experiment's control, we utilized the non-parametric Kruskal–Wallis test, which does not require a normal distribution. A P value of <0.05 was considered statistically significant if not indicated otherwise. The analysis of the RT experiments was performed with START-R suite, and Wilcoxon test was performed to evaluate the significance of the differences in RT with a P value of $<10^{-3}$ defined as significant. In all graphs, results were either shown as mean ± SD or median with a range from the 25th to 75th percentile; details for each graph are listed in the corresponding figure's legend.

### Online supplemental material

Fig. S1 shows the molecular characterization of MCF10A DBF4- and DRF1-deficient cells. Fig. S2 shows the efficiency of DBF4- and DRF1-targeting siRNAs and their effects on the rate of DNA synthesis. Fig. S3 shows regions of chromosomes 11, 17, 4, and 5 with changed RT in MCF10A cells treated with XL413 or in DBF4-deficient cells and the analysis of changes in RT for regions containing large genes, constitutive origins, CpG island, and putative G4 structures.

### Data availability

The data underlying the RT experiments in Fig. 3 and Fig. S3 are openly available in GEO with accession number GSE248981.

## Acknowledgments

We thank Lucy Drury and John Diffley, The Francis Crick Institute, London, UK, for the anti-DBF4 antibodies, Raimundo Freire, Canary Islands University Hospital Research Unit, ITB-ULL, La Laguna, Spain, for the attempts to generate anti-DRF1 antibodies, Barbara De Kegel and Colm Ryan, University College Dublin, Dublin, Ireland, for help in the computational analysis, and Sandra Healy, University of Galway, Galway, Ireland, for the critical reading of the manuscript. We thank Enda O'Connell and Shirley Hanley at the screening and genomics and flow cytometry facilities at the University of Galway. We also thank Nicolas Valentin for performing the flow cytometry cell sorting at the ImagoSeine core facility of the Institut Jacques-Monod, Paris, France.

This work was supported by Science Foundation Ireland grant 16/IA/4476 and by La Ligue Nationale Contre le Cancer (RS16/75-108 and RS17/75-135), GEFLUC, the Institut National du Cancer INCa-10493, the IdEx Université de Paris ANR-18-IDEX-0001, and by the generous legacy of Ms. Suzanne Larzat to J.-C. Cadoret's group. D.Shamavu is supported by an Irish Research Council Scholarship GOIPG/2022/896. Open access funding provided by Irish Research eLibrary.

Author contributions: A. Göder: Conceptualization, Investigation, Visualization, Writing—original draft, Writing—review and editing, C.A. Maric: Conceptualization, Data curation, Formal analysis, Investigation, Validation, Visualization, Writing—original draft, Writing—review and editing, M.D. Rainey: Formal analysis, Investigation, Methodology, Resources, Visualization, A. O'Connor: Investigation, C. Cazzaniga: Investigation, D. Shamavu: Investigation, J.-C. Cadoret: Conceptualization, Writing—review and editing, C. Santocanale: Conceptualization, Funding acquisition, Supervision, Writing—original draft, Writing—review and editing.

Disclosures: A. Göder reported personal fees from AstraZeneca outside the submitted work; and "While unrelated to the submitted work, A. Göder would like to disclose that she is currently employed by AstraZeneca and has stock ownership and/or stock options or interests in the company." C. Santocanale reported personal fees from Turbine LTD outside the submitted work; in addition, C. Santocanale had a patent to EP4275686A1 pending. No other disclosures were reported.

Submitted: 26 February 2024

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

**Supplemental material**

**A** Sequencing results of MCF10A DBF4-deficient cell lines:

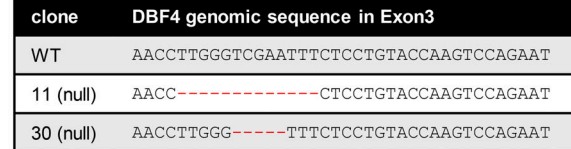

| clone | DBF4 genomic sequence in Exon3 |
|---|---|
| WT | AACCTTGGGTCGAATTTCTCCTGTACCAAGTCCAGAAT |
| 11 (null) | AACC-------------CTCCTGTACCAAGTCCAGAAT |
| 30 (null) | AACCTTGGG-----TTTCTCCTGTACCAAGTCCAGAAT |

**C**

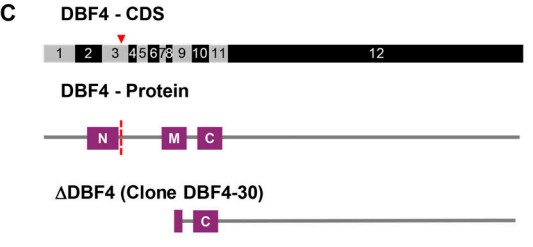

DBF4 - CDS

1 2 3 4 5 6 7 8 9 10 11 12

DBF4 - Protein

N M C

ΔDBF4 (Clone DBF4-30)

C

**B**

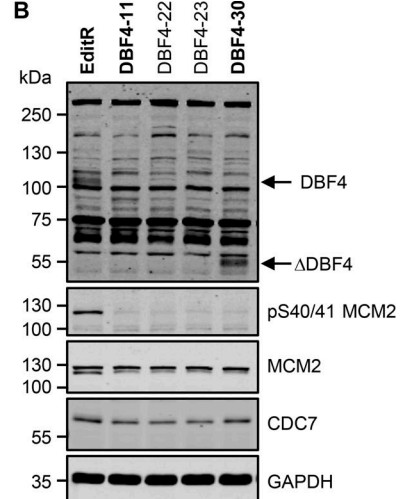

EditR   DBF4-11   DBF4-22   DBF4-23   DBF4-30

kDa
250
130
100 ← DBF4
75
55 ← ΔDBF4

130
100   pS40/41 MCM2

130
100   MCM2

55   CDC7

35   GAPDH

**D** >DBF4 WT amino acid sequence (674 aa):

MNSGAMRIHSKGHFQGGIQVKNEKNRPSLKSLKTDNRPEKSKCKPLW**GKVFYLDLPSVTISEKLQKDIKDLGGRVEEFLSKDISYLISNKKE**AKFAQ
TLGRISPVPSPESAYTAETTSPHPSHDGSSFKSPDTVCLSRGKLLVEKAIKDHDFIPSNSILSNALSWGVKILHIDDIRYYIEQKKKELYLLKKSSTSVRD
GGKRVGSGAQKTRTG**RLKKPFVKVEDMSQLYRPF**YLQLTNMPFINYSIQKPCSPFDVDKPSSMQKQTQVKLRIQTDGDKYGGTSIQLQLKEK**KKK**
**GYCECCLQKYEDLETHLLSEQHRNFAQSNQYQVVDDIV**SKLVFDFVEYEKDTPKKKRIKYSVGSLSPVSASVLKKTEQKEKVELQHISQKDCQEDD
TTVKEQNFLYKETQETEKKLLFISEPIPHPSNELRGLNEKMSNKCSMLSTAEDDIRQNFTQLPLHKNKQECILDISEHTLSENDLEELRVDHYKCNIQA
SVHVSDFSTDNSGSQPKQKSDTVLFPAKDLKEKDLHSIFTHDSGLITINSSQEHLTVQAKAPFHTPPEEPNECDFKNMDSLPSGKIHRKVKIILGRNR
KENLEPNAEFDKRTEFITQEENRICSSPVQSLLDLFQTSEEKSEFLGFTSYTEKSGICNVLDIWEEENSDNLLTAFFSSPSTSTFTGF*

>DBF4-30 Putative sequence of C-ter fragment (450 aa):

**MSQLYRPF**YLQLTNMPFINYSIQKPCSPFDVDKPSSMQKQTQVKLRIQTDGDKYGGTSIQLQLKEK**KKKGYCECCLQKYEDLETHLLSEQHRNFA**
**QSNQYQVVDDIV**SKLVFDFVEYEKDTPKKKRIKYSVGSLSPVSASVLKKTEQKEKVELQHISQKDCQEDDTTVKEQNFLYKETQETEKKLLFISEPIP
HPSNELRGLNEKMSNKCSMLSTAEDDIRQNFTQLPLHKNKQECILDISEHTLSENDLEELRVDHYKCNIQASVHVSDFSTDNSGSQPKQKSDTVLFP
AKDLKEKDLHSIFTHDSGLITINSSQEHLTVQAKAPFHTPPEEPNECDFKNMDSLPSGKIHRKVKIILGRNRKENLEPNAEFDKRTEFITQEENRICSSP
VQSLLDLFQTSEEKSEFLGFTSYTEKSGICNVLDIWEEENSDNLLTAFFSSPSTSTFTGF*

**E** Sequencing results of MCF10A DRF1-deficient cell lines:

| clone | DRF1 genomic sequence in Exon9 |
|---|---|
| WT | GGCCCCGTTCCTCAAAATCGAAGATGAAAGCAGG |
| 5 (null) | GGCC-------TCAAAATCGAAGATGAAAGCAGG |
| 7 (null) | GGCCCCGTTTCCTCAAAATCGAAGATGAAAGCAGG |

**F**

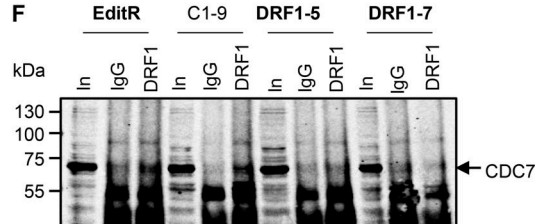

EditR   C1-9   DRF1-5   DRF1-7

kDa   In IgG DRF1   In IgG DRF1   In IgG DRF1   In IgG DRF1
130
100
75                                                      ← CDC7
55

**G**

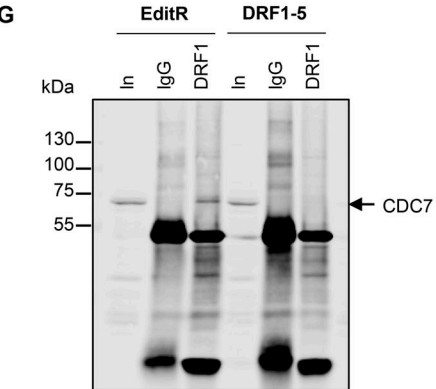

EditR   DRF1-5

kDa   In IgG DRF1   In IgG DRF1
130
100
75   ← CDC7
55

Figure S1.   **Characterization of MCF10A DBF4- and DRF1-deficient cells. (A)** Genomic DNA was extracted from MCF10A, DBF4-11, and DBF4-30 cells, and PCR products spanning *DBF4* exon 3 were sequenced. **(B)** Protein extracts prepared from MCF10A EditR, DBF4-11, and DBF4-30 (bold) as well as from two other clones that were not used in the study were analyzed by immunoblotting with the indicated antibodies. Arrows indicate full-length DBF4 and DBF4 C-terminus fragment. **(C)** Scheme of DBF4 gene and protein, indicating the position of the deletions and position where the change in protein sequence occurs, and scheme of the predicted DBF4 fragment detected in clone DBF4-30. The red triangle marks the position of the Cas9 cut site in the coding sequence (CDS) for DBF4, and the red dotted line indicates where the change in the sequence of the DBF4 protein occurs in the mutants. **(D)** Sequences of full-length DBF4 and possible translated DBF4 product expressed in the clone DBF4-30; N, M, and C motifs are underlined and in bold. **(E)** Genomic DNA was extracted from MCF10A, DRF1-5, and DRF1-7 cells, and PCR products spanning *DRF1* exon 9 were sequenced. **(F)** Protein extracts were prepared from MCF10A EditR, DRF1-5, DRF1-7 (bold), and a different clone not used in this study. IP was then performed with either anti-DRF1 mAb 5G4 or unrelated IgGs and probed with the indicated antibodies. Protein extract was loaded as control (In); arrow indicates CDC7 protein. **(G)** Protein extracts were prepared from MCF10A EditR or DRF1-5 and an independent IP experiment was performed as in panel F; arrow indicates CDC7 protein. Source data are available for this figure: SourceData FS1.

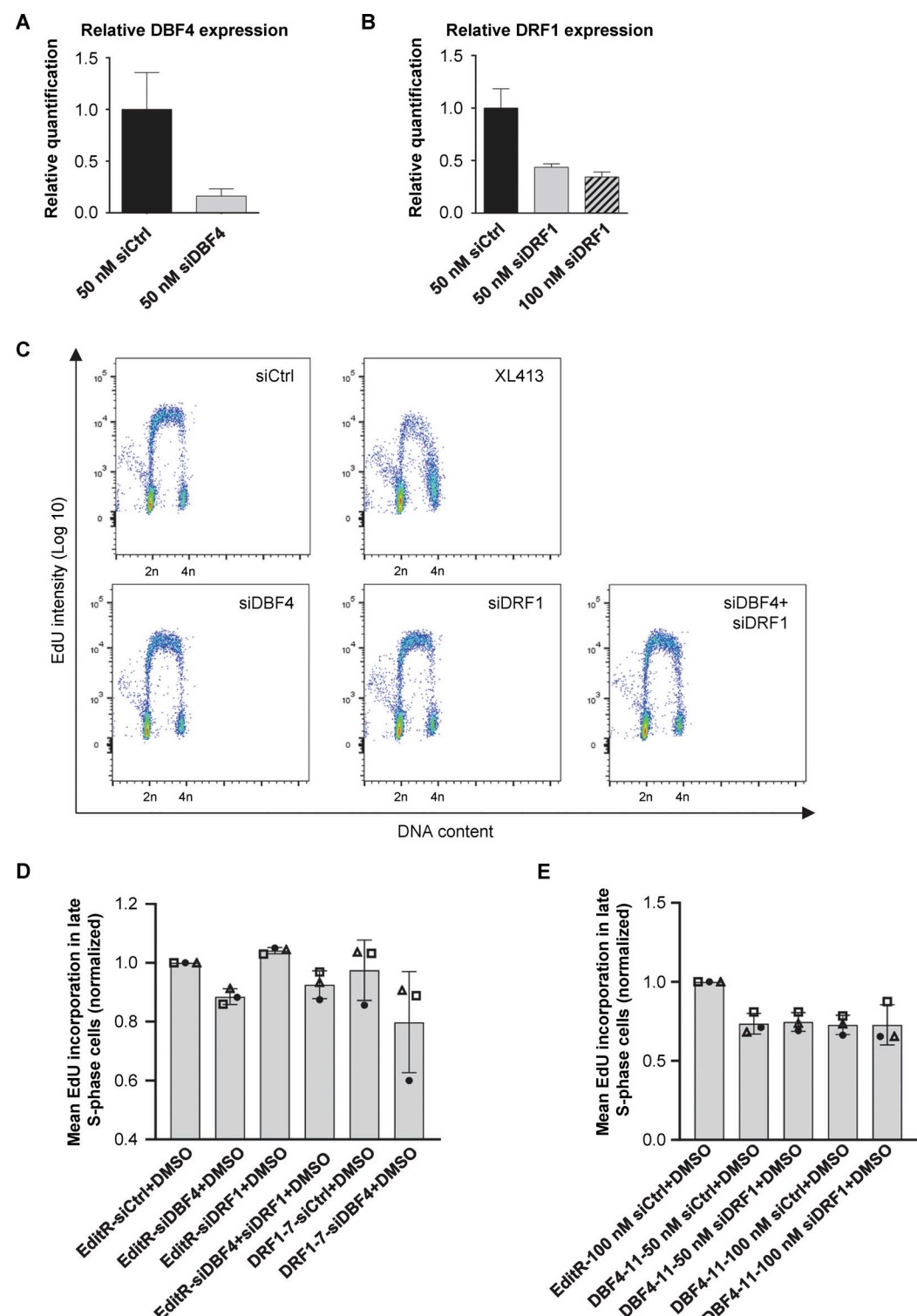

Figure S2.   **DBF4 depletion partially reduces DNA replication rate in MCF10A cells. (A and B)** MCF10A EditR cells were transfected with siCtrl, siDBF4 (A), or siDRF1 (B) at indicated concentrations over 72 h. DBF4 and DRF1 mRNA levels were assessed using real-time qPCR. Data are representative of one independent experiment performed in technical triplicates. **(C)** MCF10A EditR was transfected with 50 nM of indicated siRNAs for 48 h followed by treatment with 10 μM XL413 or DMSO for 24 h. For flow cytometry analysis, cells were labeled with 10 μM EdU 30 min prior to harvest. Representative images from one of three independent experiments are shown. **(D)** Analysis of fluorescence intensity, proportional to EdU incorporation, in late S-phase cells described in C and including additional samples of DRF1-7 cells transfected with 50 nM siCtrl or siDBF4 of the same experiment. Mean fluorescence intensity in late S-phase cells of three independent experiments was expressed as a ratio relative to MCF10A EditR cells transfected with siCtrl. Experiments are represented with different symbols, and columns are displayed as mean ± SDs. **(E)** Analysis of fluorescence intensity, proportional to EdU incorporation, in MCF10A EditR and DBF4-11 cells transfected and treated as indicated in late S-phase of three independent experiments. Mean fluorescence intensity was expressed as a ratio relative to MCF10A EditR cells transfected with siCtrl. Experiments are represented with different symbols, and columns are displayed as mean ± SDs.

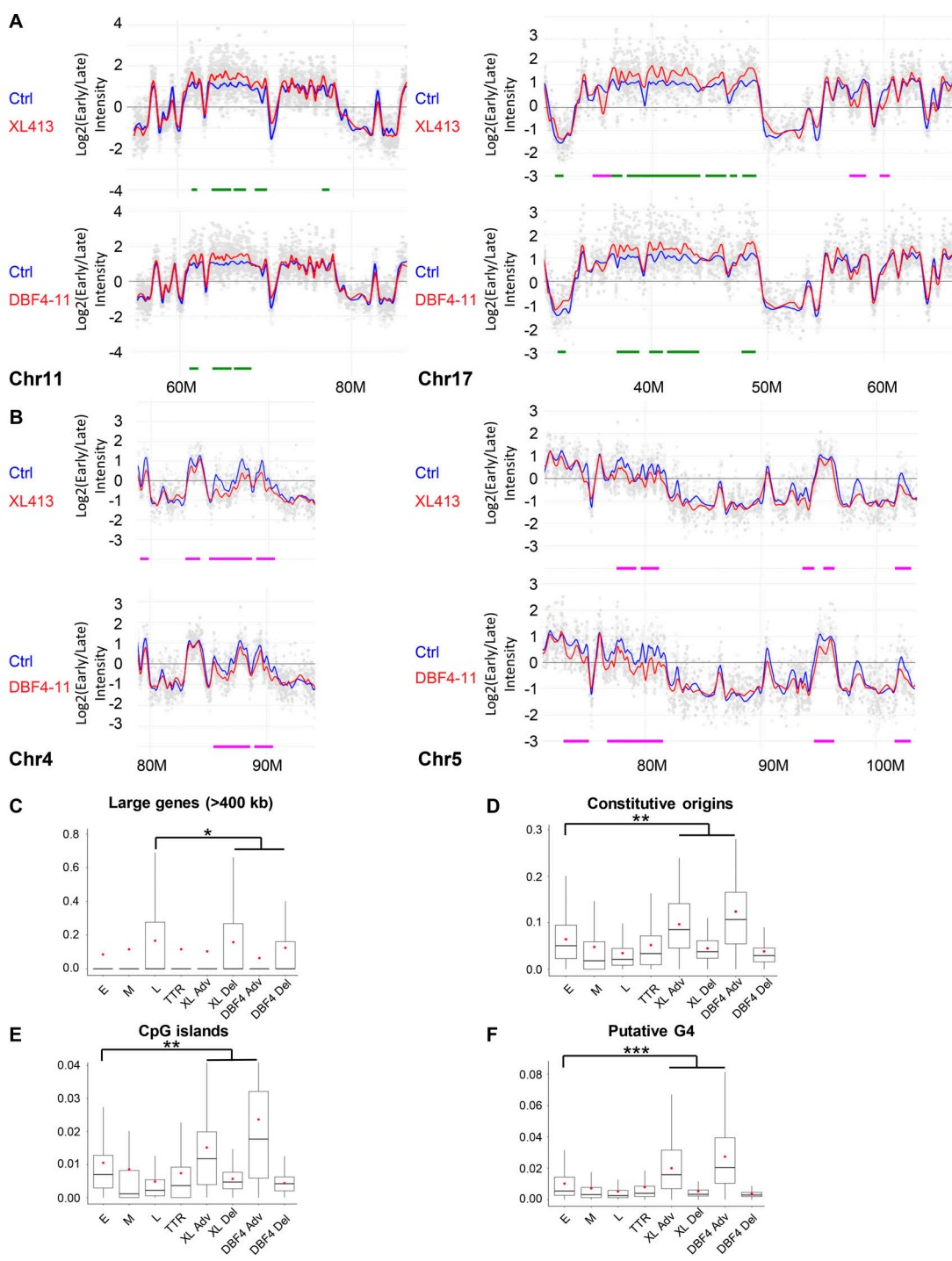

Figure S3. **Regions with changed RT in MCF10A cells treated with XL413 or in DBF4-deficient cells share similar properties. (A)** Part of chromosomes 11 and 17 RT profiles with mostly advanced regions upon CDC7 inhibition with XL413 (top profiles) or in DBF4-deficient cells (bottom profiles). RT profiles display the log ratio between early and late replicated fractions along the chromosome. Positive log ratios correspond to early replicated regions whereas negative ones correspond to late replicated regions. The blue line represents MCF10A EditR cells treated with DMSO, the red one, cells treated with 10 µM XL413 or DBF4-deficient cells (DBF4-11). Chromosome coordinates are indicated below the profile in megabases (M). Differences in RT are marked below the profiles with advanced regions in green and delayed regions in magenta. Data are representative of two replicates of four independent experiments. **(B)** Part of chromosomes 4 and 5 RT profiles with delayed regions upon CDC7 inhibition with XL413 (top profiles) or in DBF4-deficient cells (bottom profiles). Analysis was performed and graphs were generated as described in A. **(C–F)** Coverage of large genes (>400 kb; C), constitutive origins (D), CpG islands (E), and regions rich in putative G4 sequences (F) in RT changing regions of MCF10A EditR and DBF4-deficient cells treated with 10 µM XL413 or DMSO for 24 h. Results were compared to the coverage of these factors in early (E), mid (M), and late (L) replicating regions or TTR of MCF10A EditR cells. Advanced regions (Adv) and delayed regions (Del) are displayed separately. The box plots show the dispersion of the data with a range from the 25th to 75th percentile, the sample median represented by the line inside the box, and the mean by a red dot. The significance of the differences was estimated with a Wilcoxon test (*P < 10^{-3}, **P < 7.5 10^{-5}, and ***P < 7.3 10^{-13}).

