## [Peer Review File · The Journal of Cell Biology]

DBF4, not DRF1, is the crucial regulator of CDC7 kinase at replication forks

Anja Göder, Chrystelle Maric, Michael Rainey, Aisling O'Connor, Chiara Cazzaniga, Daniel Shamavu, Jean-Charles CADORET, and Corrado Santocanale

Corresponding Author(s): Corrado Santocanale, Ollscoil na Gaillimhe - University of Galway

Review Timeline:

Submission Date:	2024-02-26
Editorial Decision:	2024-03-07
Revision Received:	2024-04-02
Editorial Decision:	2024-04-29
Revision Received:	2024-05-02

Monitoring Editor: Agata Smogorzewska

Scientific Editor: Dan Simon

Transaction Report:

DOI: <https://doi.org/10.1083/jcb.202402144>

Revision 0

Review #1

1. Evidence, reproducibility and clarity:

Evidence, reproducibility and clarity (Required)

DBF4 and DRF1 knockout cells were generated and used to separate DBF4- and CDC7-dependent from DRF1- and CDC7-dependent activities. DBF4- and CDC7-dependent activities at replication forks were independent of DRF1. These include the replication timing pattern, replication fork velocity, DNA damage signaling. DBF4 is required to recruit CDC7 to active replication forks.

The study is in large part exceptional. The inclusion of quantitation for a modest bandshift on CDC7 in figure 2 (30% vs 50% reduced) is not justified given the abundance of the main band and our knowledge of the lack of linearity of western blot quantitation. This should be removed.

The only significant weakness in the paper is the explanation of the replication timing analyses in Figure 3. I don't understand what the differences between the plots equate to in terms of timing. I understand the replication of these regions that diverge is either early or late, but there were only two fractions of cells - 2N-3N and 3N-4N (the cells are "normal"). If this is the case, isn't the readout binary? a sequence either replicates in S phase between 2N and 3N or in S phase between 3N and 4N. Why are the differences so small? Are they only evident in a small population of cells? If that is the case, then what does the difference really mean? I think the description of these data needs to be precise.

2. Significance:

Significance (Required)

I think this paper is a significant advance that should be published. CDC7 is a critical kinase and identifying its co-factor at the replication fork is important both for our understanding of mechanisms of DNA replication and the impact of CDC7 kinase inhibitors in the clinic. I think the majority of the experiments are well designed and the results are unambiguous and precisely described.

3. How much time do you estimate the authors will need to complete the suggested revisions:

Estimated time to Complete Revisions (Required)

(Decision Recommendation)

Between 1 and 3 months

Yes

Review #2

1. Evidence, reproducibility and clarity:

Evidence, reproducibility and clarity (Required)

CDC7 is a master cell cycle kinase with essential functions in DNA replication and important roles in the DNA damage response. For its functions, CDC7 relies on a regulatory factor, DBF4, which is essential in many species but not in human cells as a consequence of the presence of a second DBF4-related factor, DRF1. In this work, Göder and colleagues study the relative relevance of these regulatory proteins in CDC7 roles. Their study reveals DBF4 as the major regulatory subunit both in DNA replication, DNA damage checkpoint and fork dynamics. The objective of the study is highly relevant to understand an essential cell cycle kinase with potential applications in cancer therapies, the experiments are well performed and the conclusions are "in principle" sound.

The major handicap of the study is the absence of western blots showing the elimination of DBF4 and DRF1 in the edited cell lines due to the lack of specific antibodies. The authors have generated homozygous mutations that lead to premature stop codons behind critical CDC7 domains. However, as they mention, it is not possible to fully exclude some proteins arising from internal start sites or exon skipping events with residual (functional or altered, and not necessarily residual) activity. This is not unexpected, especially for essential proteins. This would not be a major handicap if the study were focused in a specific factor because it would only question the impact of but not the affected function, but it aims to compare the relative effect of two defective genes. In this case, it is essential to confirm that both genes are

eliminated, at least to the same degree. The computational analysis in Figure 1C is consistent with the major conclusion about the primary regulatory role of DBF4 in replication, but it is insufficient to validate the specific phenotypes addressed in the study. Indeed, there is a result that is hard to understand if the edited cell lines are defective in the expression of the regulators, specially DRF1. Figure S2D-E shows no synergistic defect in DNA synthesis when the second regulator is knock down with specific siRNAs, not even DRF1 defective cell lines treated with a siDBF4 that reduces its expression 10 times. Also, it is not clear why the defects, specially in DBF4-defective cell lines, are less severe than in cells treated with an inhibitor that causes a partial inhibition of CDC7. If it is due to the expression of DRF4, a siRNA against DRF4 should cause more severe defects.

****Minor points****

- Title in Pag 12. "DBF4 mediates the majority of CDC7 functions in the replication stress response". In this section the authors address only the role of CDC7 in checkpoint signalling but not in other processes related to the replication stress response.
- Figure 2. "EdU incorporation in late S-phase/ per cell" is clearer
- Right panels in Figures 3A and 3B are duplicated

****Referees cross-commenting****

I am aware of the difficulty to sort out the detection problem, a major handicap of the work. Immunoprecipitation as suggested by rev. 3 might be an interesting possibility. The results should be published, in any case, as they are well performed and try to answer a relevant question. But, if finally the authors fail to detect the proteins, they should make clear in the paper the limitation of their conclusions by the possibility that the expression of the regulators is not completely eliminated or could be altered. Indeed, the apparent contradiction with Suski's results raised by Rev 3 might be discussed in this context. Also, it is important to explain the lack of synergism when combining the edited mutations with siRNAs.

2. Significance:

Significance (Required)

In summary, the work is relevant and interesting, but the lack of controls about the effect of the edition rises important concerns about the conclusions. It is evident from the acknowledgment section that the authors have tried without success to generate specific antibodies. An alternative possibility would be 1) to get similar results with at least two clones addressing different exons (actually, only one clone was used for DRF1 in most cases) and 2) show synergistic effects for the more important phenotypes in edited cells transfected with efficient siRNAs. This is particularly important for DRF1-defective cells, which show no phenotypes except for an increase in micronuclei. If DBF4 is not essential because the complementary activity of DRF1, impairment of DBF4 expression with siRNAs in DRF1 deficient cells should cause synergistic defects at least in DNA replication and cell viability.

3. How much time do you estimate the authors will need to complete the suggested revisions:

Estimated time to Complete Revisions (Required)

(Decision Recommendation)

More than 6 months

Yes

Review #3

1. Evidence, reproducibility and clarity:

Evidence, reproducibility and clarity (Required)

****Summary****

Assembly of the CMG helicase during DNA replication initiation is regulated by the DBF-Dependent Kinase known as CDC7 (or DDK), which also plays roles at DNA replication forks during elongation. In vertebrates, DDK has two regulatory subunits called DBF4 and DRF1. Until now, the division of labour between these two activators of CDC7 was poorly understood in mammalian cells. To address this issue, the authors used CRISPR-Cas9 to edit the DBF4 and DRF1 genes in immortalised human breast cells (MCF10A), thereby truncating key domains of the DBF4 and DRF1 proteins. The DBF4-deficient and DRF1-deficient lines are viable, whereas the double mutant was unobtainable and likely inviable, as reported previously by the authors for knockout of CDC7 in MCF10A cells. The authors compare the DBF4-deficient and DRF1-deficient lines with the CDC7 inhibitor XL413, providing evidence that DBF4 has the major role in supporting CDC7 activity in MCF10A cells compared to DRF1, in terms of DNA replication,

origin firing, fork progression, and checkpoint activation. Curiously, DRF1 appears to be more important in preventing the formation of micronuclei - another phenotype seen upon inhibition of CDC7 kinase activity.

****Major comments:****

The data are of high quality and the key conclusions are convincing, although it is unfortunate that the authors were not able to monitor the level of DBF4 and DRF1 by immunoblotting to validate their edited cell lines. The authors previously reported using immunoprecipitation of CDC7, DBF4 and DRF1 (Tenca et al, 2007, 10.1074/jbc.M604457200) to monitor DDK subunits in HeLa cells, which would presumably have been helpful here in MCF10A cells. Nevertheless, the DNA sequence of the edited clones indicates frameshift mutations that lead to premature STOP codons, and the various phenotypes reported in this manuscript are consistent with loss of DBF4 / DRF1 function as described.

****Minor comments:****

1. The authors should discuss their data in the context of the recent study by Suski et al (<https://doi.org/10.1038/s41586-022-04698>). The latter study reported that knockout of DBF4 in mouse fibroblasts impairs proliferation but is not lethal, in agreement with the present manuscript, but Suski et al also argue that CDC7 is dispensable for DNA replication in mammalian cells due to redundancy with CDK1.
2. Some discussion of the increased frequency of micronuclei in DRF1-deficient cells compared to DBF4-deficient lines would be useful (c.f. Figure 1F-G).
3. It would be helpful to present actual p values in Figure 2, rather than asterisks.

2. Significance:

Significance (Required)

The main strength of this manuscript is the exploration of the division of labour between DBF4 and DRF1 in human cells, regarding the roles of CDC7 kinase during DNA replication initiation, fork progression and checkpoint control. A limitation would be the failure to monitor the level of DBF4 and DRF1 in the CRISPR-edited cell lines, whilst it is also possible that the relative roles of DBF4 and DRF1 might vary in different cell types.

Previous studies of DNA replication in *Xenopus* egg extracts (e.g. Takahashi et al, 2005: doi: 10.1101/gad.1339805) indicated that DRF1 is the dominant activator of CDC7. In contrast, past work from the current authors (Tenca et al, 2007, 10.1074/jbc.M604457200) indicated that DBF4 is the major partner of CDC7 in human HeLa cells, at least at the level of promoting MCM2 phosphorylation (the only parameter monitored in the previous study, whereas the present manuscript goes much deeper into the various roles of CDC7 in DNA replication control and focusses on the role of CDC7 at replication forks and in checkpoint control).

This study should be of interest to those studying chromosome replication, checkpoints and genome integrity. It should also interest those with a more clinical perspective, due to the

potential importance of CDC7 kinase inhibitors as anti-cancer agents.

My own expertise is in the field of chromosome replication.

3. How much time do you estimate the authors will need to complete the suggested revisions:

Estimated time to Complete Revisions (Required)

(Decision Recommendation)

Between 1 and 3 months

No

Reviewers' comments

Reviewer #1 (Evidence, reproducibility and clarity (Required)):

DBF4 and DRF1 knockout cells were generated and used to separate DBF4- and CDC7-dependent from DRF1- and CDC7-dependent activities. DBF4- and CDC7-dependent activities at replication forks were independent of DRF1. These include the replication timing pattern, replication fork velocity, DNA damage signaling. DBF4 is required to recruit CDC7 to active replication forks.

The study is in large part exceptional.

The inclusion of quantitation for a modest bandshift on CDC7 in figure 2 (30% vs 50% reduced) is not justified given the abundance of the main band and our knowledge of the lack of linearity of western blot quantitation. This should be removed.

>We thank the reviewer for evaluating our manuscript and for the positive feedback. In the revised manuscript we have removed the quantification of the bandshift related to CDC7 autophosphorylation in mitotic cells which was reported in Figure 1E. We recognise that the quantification may not be accurate although performed using semiquantitative near-infrared scanning technology. Importantly the experiment was performed three times with almost identical results.

The only significant weakness in the paper is the explanation of the replication timing analyses in Figure 3. I don't understand what the differences between the plots equate to in terms of timing. I understand the replication of these regions that diverge is either early or late, but there were only two fractions of cells - 2N-3N and 3N-4N (the cells are "normal"). If this is the case, isn't the readout binary? a sequence either replicates in S phase between 2N and 3N or in S phase between 3N and 4N. Why are the differences so small? Are they only evident in a small population of cells? If that is the case, then what does the difference really mean? I think the description of these data needs to be precise.

> The replication timing experiments were performed with a well-established and reliable protocol (Ryba et al., 2011, <https://doi.org/10.1038/nprot.2011.328>). Asynchronous cells are labelled with a short pulse of BrdU, and sorted in two fractions, early and late S-phase, as described in Hiratani et al., 2008, Ryba et al., 2010, Hadjadj et al., 2016 and 2020 (<https://doi.org/10.1371/journal.pbio.0060245>) (<https://doi.org/10.1101/gr.099655.109>, <https://doi.org/10.1016/j.gdata.2016.07.003>, <https://doi.org/10.1093/nargab/lqaa045>).

This method does not take into account the variation in the DNA copy number (2N vs 4N) between replicated and non-replicated parts of the genome (S/G1 ratio) as in Siefert et al., 2017 (<https://doi.org/10.1101/gr.218602.116>).

The profiles depict the average replication timing of a population of 20,000,000 cells; thus, the readout is not binary.

Replication timing profiles display the log ratio between early and late replicated fractions along the chromosome. Early replicated regions show positive log ratios and late replicated regions show negative ratios. The differential analysis performed with the START-R suite

allows the comparison of the profiles (Ctrl vs either CDC7i-treated or DBF4-deficient cells). The genomic regions with altered timing are shown in green or in purple below the profiles, showing advanced and delayed regions, respectively.

Importantly, the differences in replication timing are expressed with log ratio, that explains why the profiles are varying from -2 (very late replicating regions) and +2 (very early replicating regions). The differences we observed in Figure 3 are representative of two experiments, each composed of two technical replicates that are highly reproducible.

To better describe the data, we have modified the text in the results section with the words in bold, as below:

“These two neo-synthesized DNA **fractions were then** hybridised on human whole genome microarrays, as previously described.

The log ratio between early and late replicated fractions was calculated and visualised for the whole genome.”

We also changed the labelling of the replication profiles in Figure 3 and former Figure S3 (now Figure S4) by adding **Log2 (Early/Late)** to intensity and added two new sentences to the figure legend 3.

“Replication timing profiles display the log ratio between early and late replicated fractions along the chromosome. Positive log ratios correspond to early replicated regions whereas negative ratios correspond to late replicated regions.”

Reviewer #1 (Significance (Required)):

I think this paper is a significant advance that should be published. CDC7 is a critical kinase and identifying its co-factor at the replication fork is important both for our understanding of mechanisms of DNA replication and the impact of CDC7 kinase inhibitors in the clinic. I think the majority of the experiments are well designed and the results are unambiguous and precisely described.

Reviewer #2 (Evidence, reproducibility and clarity (Required)):

CDC7 is a master cell cycle kinase with essential functions in DNA replication and important roles in the DNA damage response. For its functions, CDC7 relies on a regulatory factor, DBF4, which is essential in many species but not in human cells as a consequence of the presence of a second DBF4-related factor, DRF1. In this work, Göder and colleagues study the relative relevance of these regulatory proteins in CDC7 roles. Their study reveals DBF4 as the major regulatory subunit both in DNA replication, DNA damage checkpoint and fork dynamics. The objective of the study is highly relevant to understand an essential cell cycle kinase with potential applications in cancer therapies, the experiments are well performed and the conclusions are "in principle" sound.

>We thank this reviewer for the time and attention in evaluating the manuscript, for the positive feedback and for indicating key points for improvement and discussion.

The major handicap of the study is the absence of western blots showing the elimination of DBF4 and DRF1 in the edited cell lines due to the lack of specific antibodies. The authors have generated homozygous mutations that lead to premature stop codons behind critical CDC7 domains. However, as they mention, it is not possible to fully exclude some proteins arising from internal start sites or exon skipping events with residual (functional or altered, and not necessarily residual) activity. This is not unexpected, especially for essential proteins. This would not be a major handicap if the study were focused in a specific factor because it would only question the impact of but not the affected function, but it aims to compare the relative effect of two defective genes. In this case, it is essential to confirm that both genes are eliminated, at least to the same degree.

> We agree with the reviewer that it would be valuable to confirm the effect of the mutations by immunoblotting.

Over the years we have had multiple attempts at generating sensitive antibodies against both DBF4 and DRF1, using recombinant proteins and synthetic peptides. We also tested several commercially available anti-DBF4 and anti-DRF1 antibodies. While often we were able to detect overexpressed proteins, the detection of endogenous levels has been particularly challenging especially in non-transformed cells, such MCF10A.

Nevertheless, with an anti-DBF4 serum we obtained from the Diffley lab, which was generated against the C-terminus fragment of hDBF4, we managed to detect endogenous full length DBF4 in parental but not in the DBF4-KO cells (this blot is now included as supplementary Fig S1B). Even with this reagent the detection levels are low and multiple non-specific immunoreactive bands are present, making the detection of DBF4 particularly challenging across the experiments. Interestingly, while DBF4 is no longer detectable in DBF4-11, one the two clones used in this work , we detect a new immunoreactive band of approximately 55kDa in the other clone DBF4-30. We reckon that this may be the result of mRNA translation from the next downstream methionine. In this case this aberrant protein would lack the N domain and most of the M domain, involved in CDC7 binding and activation, and thus this fragment is very likely not functional.

Importantly, most results in this study were obtained using both DBF4-11 and DBF4-30 clones with indistinguishable results. Only the replication timing experiments were done using a single clone DBF4-11, in which DBF4 protein is not detected.

We had less success with the direct detection of DRF1. As also suggested by reviewer #3, to screen the clones after genome editing, we originally performed IP-western experiments. We used an anti-DRF1 mAb and unrelated IgG for the immunoprecipitations and an anti-CDC7 antibody as a probe in western blotting. We detected an immunoreactive band above the background at the expected molecular weight for CDC7 when the immunoprecipitation was performed with extracts from parental cells (as well as in a clone obtained with a different sgRNA, targeting DRF1 Exon1 and never used in this study) but not when the immunoprecipitation was performed with extracts from the DRF1-5 and DRF1-7 clones used in the study. These original co-IPs are credible although not particularly pretty and importantly the result was confirmed in a more convincing experiment in the DRF1-5 clone. These new data are now included in the resubmission in Figure S1.

So, while the detection of the CDC7 regulatory subunits still remains particularly difficult, we can now provide evidence that their expression is altered in the engineered cell lines used in the study.

The computational analysis in Figure 1C is consistent with the major conclusion about the primary regulatory role of DBF4 in replication, but it is insufficient to validate the specific phenotypes addressed in the study.

> The figure reports the effects of targeting single genes with multiple sgRNA (4 to 8 according to the library used) on proliferation rate/fitness measured after multiple days in more than 1000 screens across many different human cell types. Loss of fitness can be due either to a direct problem with DNA replication or with other cellular processes.

We agree with the reviewer that the analysis in Fig 1C is consistent with the phenotypes shown in the study. Particularly it is consistent with the lack of a major defect of DRF1-deficient cells in DNA replication, and it strongly indicates an essential role for CDC7 which was somehow challenged by Suski and co-workers (see also below)

Indeed, there is a result that is hard to understand if the edited cell lines are defective in the expression of the regulators, specially DRF1. Figure S2D-E shows no synergistic defect in DNA synthesis when the second regulator is knock down with specific siRNAs, not even DRF1 defective cell lines treated with a siDBF4 that reduces its expression 10 times. Also, it is not clear why the defects, specially in DBF4-defective cell lines, are less severe than in cells treated with an inhibitor that causes a partial inhibition of CDC7. If it is due to the expression of DRF4, a siRNA against DRF4 should cause more severe defects.

> Yes, we did not detect synergy or additive effect on the rate of DNA replication when targeting both DBF4 and DRF1 by multiple approaches. This was also for us an unexpected result, that we examined to the best of our capabilities.

The lack of the expected synergy in the replication assays could be explained in multiple ways and could be of biological or technical nature such as 1) residual low levels of

DBF4/DRF1 proteins remaining in the cells upon either CRISPR/Cas9 or siRNA targeting, 2) alternative mechanisms of kinase activation by a different, yet unidentified protein, 3) minimal residual enzymatic activity of hCdc7 kinase not requiring an activating subunit.

We performed further computational analysis using the dataset of the DepMap project, assessing if the effect of targeting DBF4 on fitness may be dependent on the levels of DRF1 expression. In several instances, when dealing with paralogues the gene effect of knocking out one of the paralogues directly correlates with the expression levels of the second, a phenomenon known as paralogue buffering (De Kegel et al. 2019 <https://doi.org/10.1371/journal.pgen.1008466>)

In the case of DBF4 and DRF1, this correlation is minimal (plot below: X and Y axes are DRF1 expression levels and DBF4 gene effect respectively, Pearson's correlation = 0.12) so that there are ~ 470 other genes whose expression is more correlated with DBF4 essentiality.

Furthermore, by stratifying cell lines according to whether DBF4 was essential or not and then looking at DBF4B (DRF1) expression, we failed to see significant association (graph below).

Thus, this analysis reinforces the idea that if cooperation between DBF4 and DRF1 exists, it is particularly difficult to demonstrate. To date the interplay between DBF4 and DRF1 is only indicated by the partial impairment on MCM2 phosphorylation and CDC7 autophosphorylation observed in the individual KOs and by the fact that we were unable to

obtaining viable double KO mutant clones. We recognise that the latter is a negative result and double KO may be generated in other cellular models or with different strategies.

We are happy to include the above computational analysis in a revised manuscript and to expand the discussion on the essentiality of CDC7, DBF4 and DRF1.

The effects of directly inhibiting CDC7 with 10 μ M XL413 (concentration used in this study) are indeed stronger than DBF4 KO / depletion on both DNA synthesis (Fig 2A-B) and MCM2 phosphorylation (Fig 4A and Fig 5A).

We and others have previously shown that CDC7 inhibition by XL413 causes a dose dependent decrease in MCM2 phosphorylation and DNA synthesis. Importantly in the experiments where XL413 was titrated on MCF10A cells from 0.3 μ M to 80 μ M, we demonstrated that these parameters are uncoupled and that doses that are ~20-fold higher are required to cause a strong impediment of DNA synthesis compared to the dose required to cause full MCM2 dephosphorylation (Rainey et al. 2017 <https://doi.org/10.1021/acscchembio.7b00117>).

DBF4 deficiency only partially affects MCM2 phosphorylation thus it is comparable to very low doses of XL413, that we can estimate to be in the range between 1 and 2 μ M.

Minor points

- Title in Pag 12. "DBF4 mediates the majority of CDC7 functions in the replication stress response". In this section the authors address only the role of CDC7 in checkpoint signalling but not in other processes related to the replication stress response.

> We agree and we have modified the title of this section accordingly.

- Figure 2. "EdU incorporation in late S-phase/ per cell" is clearer

> We have modified the label of this figure.

- Right panels in Figures 3A and 3B are duplicated

> We sincerely apologise for the mistake occurred while assembling the figure. The figure has been corrected, and shows that the changes in the replication timing with the CDC7i or with DBF4-KO are indeed similar but not identical.

****Referees cross-commenting****

I am aware of the difficulty to sort out the detection problem, a major handicap of the work. Immunoprecipitation as suggested by rev. 3 might be an interesting possibility. The results should be published, in any case, as they are well performed and try to answer a relevant question. But, if finally the authors fail to detect the proteins, they should make clear in the paper the limitation of their conclusions by the possibility that the expression of the

regulators is not completely eliminated or could be altered. Indeed, the apparent contradiction with Suski's results raised by Rev 3 might be discussed in this context.

>We appreciate the reviewer's recognition of the technical problems we have encountered. We are glad that we now are in a position to provide evidence of impairment of DBF4 and DRF1 expression in the engineered cells (discussed above and reported in new Figure S1 and S2).

Also, it is important to explain the lack of synergism when combining the edited mutations with siRNAs.

> In a revised manuscript we will explain the potential reasons why lack of synergism either doesn't exist or is not observed, as discussed above.

Reviewer #2 (Significance (Required)):

In summary, the work is relevant and interesting, but the lack of controls about the effect of the edition rises important concerns about the conclusions. It is evident from the acknowledgment section that the authors have tried without success to generate specific antibodies. An alternative possibility would be 1) to get similar results with at least two clones addressing different exons (actually, only one clone was used for DRF1 in most cases) and 2) show synergistic effects for the more important phenotypes in edited cells transfected with efficient siRNAs. This is particularly important for DRF1-defective cells, which show no phenotypes except for an increase in micronuclei. If DBF4 is not essential because the complementary activity of DRF1, impairment of DBF4 expression with siRNAs in DRF1 deficient cells should cause synergistic defects at least in DNA replication and cell viability.

> We hope we have satisfactory addressed this reviewer's comments, by providing experimental evidence of the impairment of DBF4 and DRF1 expression/function in the engineered cells and several points for discussion addressing the lack of obvious synergy between DBF4 and DRF1.

Reviewer #3 (Evidence, reproducibility and clarity (Required)):

Summary

Assembly of the CMG helicase during DNA replication initiation is regulated by the DBF-Dependent Kinase known as CDC7 (or DDK), which also plays roles at DNA replication forks during elongation. In vertebrates, DDK has two regulatory subunits called DBF4 and DRF1. Until now, the division of labour between these two activators of CDC7 was poorly understood in mammalian cells. To address this issue, the authors used CRISPR-Cas9 to edit the DBF4 and DRF1 genes in immortalised human breast cells (MCF10A), thereby truncating key domains of the DBF4 and DRF1 proteins. The DBF4-deficient and DRF1-deficient lines are viable, whereas the double mutant was unobtainable and likely inviable, as reported previously by the authors for knockout of CDC7 in MCF10A cells. The authors compare the DBF4-deficient and DRF1-deficient lines with the CDC7 inhibitor XL413, providing evidence that DBF4 has the major role in supporting CDC7 activity in MCF10A cells compared to DRF1, in terms of DNA replication, origin firing, fork progression, and checkpoint activation. Curiously, DRF1 appears to be more important in preventing the formation of micronuclei - another phenotype seen upon inhibition of CDC7 kinase activity.

Major comments:

The data are of high quality and the key conclusions are convincing, although it is unfortunate that the authors were not able to monitor the level of DBF4 and DRF1 by immunoblotting to validate their edited cell lines. The authors previously reported using immunoprecipitation of CDC7, DBF4 and DRF1 (Tenca et al, 2007, 10.1074/jbc.M604457200) to monitor DDK subunits in HeLa cells, which would presumably have been helpful here in MCF10A cells. Nevertheless, the DNA sequence of the edited clones indicates frameshift mutations that lead to premature STOP codons, and the various phenotypes reported in this manuscript are consistent with loss of DBF4 / DRF1 function as described.

> We thank the reviewer the time an effort in carefully assessing the manuscript, and with his/her positive assessment.

We have now included experimental evidence indicating that DBF4 expression is deficient in the DBF4 KO cells used in this study and that the interaction with DRF1 and CDC7 is deficient in the DRF1-KO cells using the same Co-IP strategy previously reported in Hela cells. Please see also the response to reviewer #2 to the same point.

Minor comments:

1. The authors should discuss their data in the context of the recent study by Suski et al (<https://doi.org/10.1038/s41586-022-04698>). The latter study reported that knockout of DBF4 in mouse fibroblasts impairs proliferation but is not lethal, in agreement with the present manuscript, but Suski et al also argue that CDC7 is dispensable for DNA replication in mammalian cells due to redundancy with CDK1.

> The requirement for CDC7 kinase activity for genome duplication in mammalian cells has become a contentious point of debate. CRISPR screens in more than 1000 cell lines indicate

that CDC7 is a core essential gene required for proliferation (DepMap.org). Clearly human cells can clearly withstand reduced CDC7 activity, and several proteins contribute both positively and negatively to the effectiveness of CDC7 inhibition in DNA replication and cell proliferation e.g. RIF1 depletion, ATR inhibition, PTBP1 mutation. (Hiraga et al. 2017 <https://doi.org/10.15252/embr.201641983> ; Rainey et al. 2020 <https://doi.org/10.1016/j.celrep.2020.108096> ; Jones et al. 2021 <https://doi.org/10.1016/j.molcel.2021.01.004> ; Göder et al. 2023 <https://doi.org/10.1016/j.isci.2023.106951>).

Specifically CDK1-phosphorylation of RIF1 was shown to disrupt RIF1/PP1 interaction and PP1's ability to counteract CDC7-dependent phosphorylation of the MCM complex (Moiseeva et al. 2019 <https://doi.org/10.1073/pnas.1903418116> ; Jones et al. 2021 <https://doi.org/10.1016/j.molcel.2021.01.004>). Thus increased CDK1 activity can be helpful in dealing with low levels of CDC7 kinase.

Suski et al argue that CDC7 is dispensable for DNA replication in human cells based on acute degradation of CDC7 or by its inhibition using an "Shokat type" analogue sensitive CDC7 allele. However, another study showed that DNA replication is not completed using the same approach and the same analogue sensitive allele (Jones et al. 2021 <https://doi.org/10.1016/j.molcel.2021.01.004>)

In mouse embryonic stem cells, the Masai group had previously shown that CRE-Lox mediated inactivation of mDBF4 leads to a strong decrease of DNA synthesis and that mDBF4, like mCDC7 is essential for cell ES cells viability (Kim et al, 2002 <https://doi.org/10.1093/emboj/21.9.2168> and Yamashita 2005 <https://doi.org/10.1111/j.1365-2443.2005.00857.x>). Intriguingly mDRF1 has yet not been identified nor characterised.

In our opinion, the simplest explanation to reconcile the different reports is that human and mouse CDC7 are indeed required for DNA replication and for cell proliferation, but the phenotype of the most severe effects of its inhibition requires the complete loss of function of the kinase and may be delayed in time.

We are happy to add these considerations in the discussion section of the revised manuscript.

2. Some discussion of the increased frequency of micronuclei in DRF1-deficient cells compared to DBF4-deficient lines would be useful (c.f. Figure 1F-G).

> In the discussion we have suggested that the increase of micronucleated cells in the DRF1 deficient clones "could be consistent with a (DRF1) specific but not yet identified function in chromosome segregation, in the fine-tuning of DNA replication or the DNA repair process". Of interest, CDC7 kinase was recently involved in modulating ATR function in cytokinetic abscission, and impairment of this process can lead to increase frequency of micronucleated cells (Luessing et al. 2023 <https://doi.org/10.1016/j.isci.2022.104536>). It is possible that this new role of CDC7 could be dependent on DRF1, an hypothesis at present purely speculative, that we will be testing in the future.

We are happy to add these considerations to the discussion section of the revised manuscript.

3. It would be helpful to present actual p values in Figure 2, rather than asterisks.

> Asterisks report the range in which the p values fall into, which currently is specified in the legend. These can be substituted with actual numbers in the figures, and we will comply with the requirement of the journal in which the manuscript will be accepted.

Reviewer #3 (Significance (Required)):

The main strength of this manuscript is the exploration of the division of labour between DBF4 and DRF1 in human cells, regarding the roles of CDC7 kinase during DNA replication initiation, fork progression and checkpoint control. A limitation would be the failure to monitor the level of DBF4 and DRF1 in the CRISPR-edited cell lines, whilst it is also possible that the relative roles of DBF4 and DRF1 might vary in different cell types.

Previous studies of DNA replication in *Xenopus* egg extracts (e.g. Takahashi et al, 2005: doi: 10.1101/gad.1339805) indicated that DRF1 is the dominant activator of CDC7. In contrast, past work from the current authors (Tenca et al, 2007, 10.1074/jbc.M604457200) indicated that DBF4 is the major partner of CDC7 in human HeLa cells, at least at the level of promoting MCM2 phosphorylation (the only parameter monitored in the previous study, whereas the present manuscript goes much deeper into the various roles of CDC7 in DNA replication control and focusses on the role of CDC7 at replication forks and in checkpoint control).

This study should be of interest to those studying chromosome replication, checkpoints and genome integrity. It should also interest those with a more clinical perspective, due to the potential importance of CDC7 kinase inhibitors as anti-cancer agents.

My own expertise is in the field of chromosome replication.

Revision Plan

Manuscript number: RC-2023-02286

Corresponding author(s): Corrado Santocanale

1. General Statements

Dear Editor ,

Please find the manuscript with title “DBF4, not DRF1, is the crucial regulator of CDC7 kinase at replication forks” which was submitted for review to Review Commons.

We are particularly gratified with the comments of the reviewers which find this manuscript to be a significant advance for the field, with experiments well designed and performed, even suggesting that the “study is in large part exceptional”.

We believe that we have addressed all the reviewers comments and further revision will be limited and focused around the discussion of our results in the context of recent and controversial literature challenging the essentiality of CDC7 and DBF4 in DNA replication and cell proliferation (Suski et al. 2022 Nature <https://doi.org/10.1038/s41586-022-04698-x>).

We will also include more discussion on why we might have failed to observe synergy between DBF4 and DRF1 and potential causes for increased frequency of micronuclei in the DRF1-deficient cells

2. Description of the planned revisions

We plan to revise the text to:

- 1) incorporate a discussion on the essentiality of CDC7 and DBF4 (reviewer #3),
- 2) explain why synergism between DBF4 and DRF1 was not observed, this may involve the inclusion of further computational analysis of the data from CRISPR screens available in DepMap portal (DepMap.org) (please see response to reviewer #2 comments),
- 3) discuss possible causes for the increased frequency of micronuclei in the DRF1-deficient cells (reviewer #3).

3. Description of the revisions that have already been incorporated in the transferred manuscript

- 1) We have removed the quantification of bandshift from figure 2B (reviewer #1).

2) We have modified the labeling of the replication profiles in Figure 3 and Figure S3 and amended figure legends; we also made minor changes in the text (results section) to better explain the procedure and the outputs of the replication timing experiments, as requested by reviewer #1.

3) We have included experimental data showing the altered expression of DBF4 and DRF1 proteins in the clones that were used in the study (comments from reviewer #2 and #3).

We have amended former Figure S1 which originally only reported sequencing data.

In the new Figure S1 panels A-D are related to DBF4 clones and report: A) sequencing data of the mutations in the clones, B) DBF4 immunoblotting in parental cells and DBF4-11, DBF4-30 as well as in two other clones that were not used in the study, C) scheme of DBF4 gene and protein, the position of the deletions and position where the change in protein sequence occurs, scheme of the predicted DBF4 fragment detected in clone DBF4-30, and D) sequences of full length DBF4 and possible translated DBF4 products in the clones.

Figure S1 panels E-I are related to DRF1 clones and report: E) sequencing data of the mutations in these clones, F) co-IP experiment in parental cells, DRF1-5, DRF1-7 and a different clone not used in this study, G) a second co-IP in parental and clone DRF1-5, H) scheme of DRF1 gene and protein, with the position of the deletions and position where the change in protein sequence occurs, and I) sequences of full length DRF1 and possible translated DRF1 products in the clones. The text in the results and methods sections has been modified to describe these additional experiments and experimental procedures.

4) We have modified the title of the section “DBF4 mediates the majority of CDC7 functions in the replication stress response” to “DBF4 mediates the majority of CDC7 functions in checkpoint signaling” (reviewer #2).

5) We have modified the labelling of Fig 2B to “EdU incorporation in late S-phase / cell” (reviewer #2).

6) We have corrected Figure 3A which now reports the correct information of replication timing in XL413 treated cells (reviewer #2).

7) During the revision we also noticed that we had mislabeled the Figure 5B. This has been corrected. These experiments were performed with both DBF4-11 and DBF4-30 as well as both DRF1-5 and DRF1-7 cells and not solely with only DBF4-11 and DRF1-7 cells as erroneously reported. There is no impact on the interpretation of the experimental results.

4. Description of analyses that authors prefer not to carry out

We have not changed the asterisks representing p values into actual numbers, as this may be subject to journal specific requirements (reviewer #3).

March 7, 2024

Re: JCB manuscript #202402144T

Prof. Corrado Santocanale
Ollscoil na Gaillimhe - University of Galway
Centre for Chromosome Biology
Biosciences Building
Newcastle Road
Galway H91W2TY
Ireland

Dear Prof. Santocanale,

Thank you for submitting your manuscript entitled "DBF4, not DRF1, is the crucial regulator of CDC7 kinase at replication forks." We have assessed the manuscript as well as the reviews from Review Commons and we invite you to submit a revised manuscript as outlined in your revision plan. Please also include discussion of your results in light of the Suski et al and other papers that assessed the essential nature of CDC7. Please note that we will ask the original reviewers to re-evaluate the revised manuscript.

We feel that the study is best suited as a Report, a short format meant for highly novel findings of broad interest. Full formatting guidelines are available on our Instructions for Authors page, <https://jcb.rupress.org/submission-guidelines#revised>.

GENERAL GUIDELINES:

Text limits: Reports must have a single 'Results and Discussion' section. Character count for Reports is < 20,000, not including spaces. Count includes title page, abstract, introduction, results & discussion, and acknowledgments. Count does not include materials and methods, figure legends, references, tables, or supplemental legends.

Figures: Reports may have up to 5 main figures. To avoid delays in production, figures must be prepared according to the policies outlined in our Instructions to Authors, under Data Presentation. All figures in accepted manuscripts will be screened prior to publication. Figures cannot span multiple pages so please either consolidate Figure S3 into 1 page or split into 2 figures. Reports are generally allowed 3 supplemental figures. In this case we will be able to give you extra space if necessary but please try to combine the current figures if possible.

Please note that JCB now requires authors to submit Source Data used to generate figures containing gels and Western blots with all revised manuscripts. This Source Data consists of fully uncropped and unprocessed images for each gel/blot displayed in the main and supplemental figures. Since your paper includes cropped gel and/or blot images, please be sure to provide one Source Data file for each figure that contains gels and/or blots along with your revised manuscript files. File names for Source Data figures should be alphanumeric without any spaces or special characters (i.e., SourceDataF#, where F# refers to the associated main figure number or SourceDataFS# for those associated with Supplementary figures). The lanes of the gels/blots should be labeled as they are in the associated figure, the place where cropping was applied should be marked (with a box), and molecular weight/size standards should be labeled wherever possible. Source Data files will be made available to reviewers during evaluation of revised manuscripts and, if your paper is eventually published in JCB, the files will be directly linked to specific figures in the published article.

When submitting the revision, please include a cover letter addressing the reviewers' comments point by point. Please also highlight all changes in the text of the manuscript. Papers are generally considered through only one revision cycle, so any revised manuscript will likely be either accepted or rejected.

Thank you for this interesting contribution to Journal of Cell Biology. You can contact us at the journal office with any questions at cellbio@rockefeller.edu.

Sincerely,

Agata Smogorzewska, MD, PhD
Monitoring Editor
Journal of Cell Biology

Dan Simon, PhD
Scientific Editor
Journal of Cell Biology

JCB resubmission: Response to Reviewers' comments

Reviewer #1 (Evidence, reproducibility and clarity (Required)):

DBF4 and DRF1 knockout cells were generated and used to separate DBF4- and CDC7-dependent from DRF1- and CDC7-dependent activities. DBF4- and CDC7-dependent activities at replication forks were independent of DRF1. These include the replication timing pattern, replication fork velocity, DNA damage signaling. DBF4 is required to recruit CDC7 to active replication forks.

The study is in large part exceptional.

The inclusion of quantitation for a modest bandshift on CDC7 in figure 2 (30% vs 50% reduced) is not justified given the abundance of the main band and our knowledge of the lack of linearity of western blot quantitation. This should be removed.

>We thank the reviewer for evaluating our manuscript and for the positive feedback. In the revised manuscript we have removed the quantification of the band shift related to CDC7 autophosphorylation in mitotic cells which was reported in Figure 1E. We recognise that the quantification may not be accurate although performed using semiquantitative near-infrared scanning technology. Importantly, the experiment was performed three times with almost identical results.

The only significant weakness in the paper is the explanation of the replication timing analyses in Figure 3. I don't understand what the differences between the plots equate to in terms of timing. I understand the replication of these regions that diverge is either early or late, but there were only two fractions of cells - 2N-3N and 3N-4N (the cells are "normal"). If this is the case, isn't the readout binary? a sequence either replicates in S phase between 2N and 3N or in S phase between 3N and 4N. Why are the differences so small? Are they only evident in a small population of cells? If that is the case, then what does the difference really mean? I think the description of these data needs to be precise.

> The replication timing experiments were performed with a well-established and reliable protocol (Ryba et al., 2011, <https://doi.org/10.1038/nprot.2011.328>). Asynchronous cells are labelled with a short pulse of BrdU, and sorted into two fractions, early and late S-phase, as described in Hiratani et al., 2008, Ryba et al., 2010, Hadjadj et al, 2016 and 2020 (<https://doi.org/10.1371/journal.pbio.0060245>) (<https://doi.org/10.1101/gr.099655.109>, <https://doi.org/10.1016/j.gdata.2016.07.003>, <https://doi.org/10.1093/nargab/lqaa045>). This method does not take into account the variation in the DNA copy number (2N vs 4N) between replicated and non-replicated parts of the genome (S/G1 ratio) as in Siefert et al., 2017 (<https://doi.org/10.1101/gr.218602.116>).

The profiles depict the average replication timing of a population of 20,000,000 cells; thus, the readout is not binary.

Replication timing profiles display the log ratio between early and late replicated fractions along the chromosome. Early replicated regions show positive log ratios and late replicated regions show negative ratios. The differential analysis performed with the START-R suite

allows the comparison of the profiles (Ctrl vs either CDC7i-treated or DBF4-deficient cells). The genomic regions with altered timing are shown in green or in purple below the profiles, showing advanced and delayed regions, respectively.

Importantly, the differences in replication timing are expressed with log ratio, that explains why the profiles are varying from -2 (very late replicating regions) to +2 (very early replicating regions). The differences we observed in Figure 3 are representative of two experiments, each composed of two technical replicates that are highly reproducible.

To better describe the data, we have modified the text in the results section:

“The DNA of neo-synthesis in these two fractions was hybridised on whole genome microarrays thus generating differential RT profiles as previously described”.

We also changed the labelling of the replication profiles in Figure 3 and Figure S3 by adding **Log2 (Early/Late)** to intensity and added two new sentences to the legends of Figure 3 and Figure S3:

“Replication timing profiles display the log ratio between early and late replicated fractions along the chromosome. Positive log ratios correspond to early replicated regions whereas negative ratios correspond to late replicated regions.”

Reviewer #1 (Significance (Required)):

I think this paper is a significant advance that should be published. CDC7 is a critical kinase and identifying its co-factor at the replication fork is important both for our understanding of mechanisms of DNA replication and the impact of CDC7 kinase inhibitors in the clinic. I think the majority of the experiments are well designed and the results are unambiguous and precisely described.

Reviewer #2 (Evidence, reproducibility and clarity (Required)):

CDC7 is a master cell cycle kinase with essential functions in DNA replication and important roles in the DNA damage response. For its functions, CDC7 relies on a regulatory factor, DBF4, which is essential in many species but not in human cells as a consequence of the presence of a second DBF4-related factor, DRF1. In this work, Göder and colleagues study the relative relevance of these regulatory proteins in CDC7 roles. Their study reveals DBF4 as the major regulatory subunit both in DNA replication, DNA damage checkpoint and fork dynamics. The objective of the study is highly relevant to understand an essential cell cycle kinase with potential applications in cancer therapies, the experiments are well performed and the conclusions are "in principle" sound.

>We thank this reviewer for the time and attention in evaluating the manuscript, for the positive feedback and for indicating key points for improvement and discussion.

The major handicap of the study is the absence of western blots showing the elimination of DBF4 and DRF1 in the edited cell lines due to the lack of specific antibodies. The authors have generated homozygous mutations that lead to premature stop codons behind critical CDC7 domains. However, as they mention, it is not possible to fully exclude some proteins arising from internal start sites or exon skipping events with residual (functional or altered, and not necessarily residual) activity. This is not unexpected, especially for essential proteins. This would not be a major handicap if the study were focused in a specific factor because it would only question the impact of but not the affected function, but it aims to compare the relative effect of two defective genes. In this case, it is essential to confirm that both genes are eliminated, at least to the same degree.

> We agree with the reviewer that it would be valuable to confirm the effect of the mutations by immunoblotting.

Over the years we have had multiple attempts at generating sensitive antibodies against both DBF4 and DRF1, using recombinant proteins and synthetic peptides. We also tested several commercially available anti-DBF4 and anti-DRF1 antibodies. While we were often able to detect overexpressed proteins, the detection of endogenous levels has been particularly challenging especially in non-transformed cells, such MCF10A.

Nevertheless, with an anti-DBF4 serum we obtained from the Diffley lab, which was generated against the C-terminus fragment of hDBF4, we managed to detect endogenous full length DBF4 in parental but not in the DBF4-KO cells (this blot is now included as supplementary Fig S1B). Even with this reagent the detection levels are low and multiple non-specific immunoreactive bands are present, making the detection of DBF4 particularly challenging across the experiments. Interestingly, while DBF4 is no longer detectable in DBF4-11, one of the two clones used in this work, we detect a new immunoreactive band of approximately 55kDa in the other clone DBF4-30. We reckon that this may be the result of mRNA translation from the next downstream methionine. In this case this aberrant protein would lack the N domain and most of the M domain, involved in CDC7 binding and activation, and thus this fragment is very likely not functional.

Importantly, most results in this study were obtained using both DBF4-11 and DBF4-30 clones with indistinguishable results.

We had less success with the direct detection of DRF1. As also suggested by reviewer #3, to screen the clones after genome editing, we originally performed IP-western experiments. We used an anti-DRF1 mAb and unrelated IgG for the immunoprecipitations and an anti-CDC7 antibody as a probe in western blotting. We detected an immunoreactive band above the background at the expected molecular weight for CDC7 when the immunoprecipitation was performed with extracts from parental cells (as well as in a clone obtained with a different sgRNA, targeting DRF1 Exon1 and never used in this study) but not when the immunoprecipitation was performed with extracts from the DRF1-5 and DRF1-7 clones used in the study. These original co-IPs are credible although not particularly pretty and importantly the result was confirmed in a more convincing experiment in the DRF1-5 clone. These new data are now included in the resubmission in Figure S1F-G.

So, while the detection of the CDC7 regulatory subunits still remains particularly difficult, we can now provide evidence that their expression is altered in the engineered cell lines used in the study.

The computational analysis in Figure 1C is consistent with the major conclusion about the primary regulatory role of DBF4 in replication, but it is insufficient to validate the specific phenotypes addressed in the study.

> The figure reports the effects of targeting single genes with multiple sgRNA (4 to 8 according to the library used) on proliferation rate/fitness measured after multiple days in more than 1000 screens across many different human cell types. Loss of fitness can be due either to a direct problem with DNA replication or with other cellular processes. We agree with the reviewer that the analysis in Fig 1C is consistent with the phenotypes shown in the study. Particularly it is consistent with the lack of a major defect of DRF1-deficient cells in DNA replication, and it strongly indicates an essential role for CDC7 which was somehow challenged by Suski and co-workers (see also below and response to reviewer #3).

Indeed, there is a result that is hard to understand if the edited cell lines are defective in the expression of the regulators, specially DRF1. Figure S2D-E shows no synergistic defect in DNA synthesis when the second regulator is knock down with specific siRNAs, not even DRF1 defective cell lines treated with a siDBF4 that reduces its expression 10 times. Also, it is not clear why the defects, specially in DBF4-defective cell lines, are less severe than in cells treated with an inhibitor that causes a partial inhibition of CDC7. If it is due to the expression of DRF4, a siRNA against DRF4 should cause more severe defects.

> Yes, we did not detect synergy or an additive effect on the rate of DNA replication when targeting both DBF4 and DRF1 by multiple approaches. This was also for us an unexpected result, that we examined to the best of our capabilities.

The lack of the expected synergy in the replication assays could be explained in multiple ways and could be of biological or technical nature such as 1) residual low levels of

DBF4/DRF1 proteins remaining in the cells upon either CRISPR/Cas9 or siRNA targeting, 2) alternative mechanisms of kinase activation by a different, yet unidentified protein, 3) minimal residual enzymatic activity of hCdc7 kinase not requiring an activating subunit. These considerations have been included in the revised manuscript.

We performed further computational analysis using the dataset of the DepMap project, assessing if the effect of targeting DBF4 on fitness may be dependent on the levels of DRF1 expression. In several instances, when dealing with paralogues the gene effect of knocking out one of the paralogues directly correlates with the expression levels of the second, a phenomenon known as paralogue buffering (De Kegel et al. 2019 <https://doi.org/10.1371/journal.pgen.1008466>)

In the case of DBF4 and DRF1, this correlation is minimal (plot below: X and Y axes are DRF1 expression levels and DBF4 gene effect respectively, Pearson's correlation = 0.12) so that there are ~ 470 other genes whose expression is more correlated with DBF4 essentiality.

Furthermore, by stratifying cell lines according to whether DBF4 was essential or not and then looking at DBF4B (DRF1) expression, we failed to see significant association (graph below).

Thus, this analysis reinforces the idea that if cooperation between DBF4 and DRF1 exists, it is particularly difficult to demonstrate. To date the interplay between DBF4 and DRF1 is only indicated by the partial impairment on MCM2 phosphorylation and CDC7

autophosphorylation observed in the individual KOs and by the fact that we were unable to obtain viable double KO mutant clones. We recognise that the latter is a negative result and double KO may be generated in other cellular models or with different strategies.

We have not included the above computational analysis in the revised manuscript due to space limitations.

The effects of directly inhibiting CDC7 with 10 μM XL413 (concentration used in this study) are indeed stronger than DBF4 KO / depletion on both DNA synthesis (Fig 2A-B) and MCM2 phosphorylation (Fig 4A and Fig 5A).

We and others have previously shown that CDC7 inhibition by XL413 causes a dose dependent decrease in MCM2 phosphorylation and DNA synthesis. Importantly in the experiments where XL413 was titrated on MCF10A cells from 0.3 μM to 80 μM , we demonstrated that these parameters are uncoupled and that doses that are ~ 20 -fold higher are required to cause a strong impediment of DNA synthesis compared to the dose required to cause full MCM2 dephosphorylation (Rainey et al. 2017 <https://doi.org/10.1021/acscchembio.7b00117>).

DBF4 deficiency only partially affects MCM2 phosphorylation thus it is comparable to very low doses of XL413, that we can estimate to be in the range between 1 and 2 μM .

Minor points

- Title in Pag 12. "DBF4 mediates the majority of CDC7 functions in the replication stress response". In this section the authors address only the role of CDC7 in checkpoint signalling but not in other processes related to the replication stress response.

> in the revised manuscript this section was merged with the section discussing the role of DBF4 at forks under a single heading: "DBF4 is required for checkpoint signalling and CDC7 activity at stalled forks"

- Figure 2. "EdU incorporation in late S-phase/ per cell" is clearer

> We have modified the label of this figure.

- Right panels in Figures 3A and 3B are duplicated

> We sincerely apologise for the mistake occurred while assembling the figure. The figure has been corrected, and shows that the changes in the replication timing with the CDC7i or with DBF4-KO are indeed similar but not identical.

****Referees cross-commenting****

I am aware of the difficulty to sort out the detection problem, a major handicap of the work. Immunoprecipitation as suggested by rev. 3 might be an interesting possibility. The results

should be published, in any case, as they are well performed and try to answer a relevant question. But, if finally the authors fail to detect the proteins, they should make clear in the paper the limitation of their conclusions by the possibility that the expression of the regulators is not completely eliminated or could be altered. Indeed, the apparent contradiction with Suski's results raised by Rev 3 might be discussed in this context.

>We appreciate the reviewer's recognition of the technical problems we have encountered. We are glad that we are now able to provide evidence of impairment of DBF4 and DRF1 expression in the engineered cells (discussed above and reported in new Figure S1).

Also, it is important to explain the lack of synergism when combining the edited mutations with siRNAs.

> In a revised manuscript we have discussed the potential reasons why lack of synergism by adding the following paragraph:

“Unexpectedly, despite the basic redundancy between DBF4 and DRF1, we did not detect synergy in reducing the rate of DNA replication when targeting both DBF4 and DRF1 by multiple approaches. The lack of synergy could be of biological or technical nature such as 1) residual low levels of DBF4/DRF1 proteins remaining in the cells, 2) alternative yet unidentified mechanisms of kinase activation, 3) minimal residual enzymatic activity of hCdc7 kinase not requiring an activating subunit. Further work will be required to test these hypotheses.”

Reviewer #2 (Significance (Required)):

In summary, the work is relevant and interesting, but the lack of controls about the effect of the edition rises important concerns about the conclusions. It is evident from the acknowledgment section that the authors have tried without success to generate specific antibodies. An alternative possibility would be 1) to get similar results with at least two clones addressing different exons (actually, only one clone was used for DRF1 in most cases) and 2) show synergistic effects for the more important phenotypes in edited cells transfected with efficient siRNAs. This is particularly important for DRF1-defective cells, which show no phenotypes except for an increase in micronuclei. If DBF4 is not essential because the complementary activity of DRF1, impairment of DBF4 expression with siRNAs in DRF1 deficient cells should cause synergistic defects at least in DNA replication and cell viability.

> We hope we have satisfactory addressed this reviewer's comments, by providing experimental evidence of the impairment of DBF4 and DRF1 expression/function in the engineered cells and several points for discussion addressing the lack of obvious synergy between DBF4 and DRF1.

Reviewer #3 (Evidence, reproducibility and clarity (Required)):

Summary

Assembly of the CMG helicase during DNA replication initiation is regulated by the DBF-Dependent Kinase known as CDC7 (or DDK), which also plays roles at DNA replication forks during elongation. In vertebrates, DDK has two regulatory subunits called DBF4 and DRF1. Until now, the division of labour between these two activators of CDC7 was poorly understood in mammalian cells. To address this issue, the authors used CRISPR-Cas9 to edit the DBF4 and DRF1 genes in immortalised human breast cells (MCF10A), thereby truncating key domains of the DBF4 and DRF1 proteins. The DBF4-deficient and DRF1-deficient lines are viable, whereas the double mutant was unobtainable and likely inviable, as reported previously by the authors for knockout of CDC7 in MCF10A cells. The authors compare the DBF4-deficient and DRF1-deficient lines with the CDC7 inhibitor XL413, providing evidence that DBF4 has the major role in supporting CDC7 activity in MCF10A cells compared to DRF1, in terms of DNA replication, origin firing, fork progression, and checkpoint activation. Curiously, DRF1 appears to be more important in preventing the formation of micronuclei - another phenotype seen upon inhibition of CDC7 kinase activity.

Major comments:

The data are of high quality and the key conclusions are convincing, although it is unfortunate that the authors were not able to monitor the level of DBF4 and DRF1 by immunoblotting to validate their edited cell lines. The authors previously reported using immunoprecipitation of CDC7, DBF4 and DRF1 (Tenca et al, 2007, 10.1074/jbc.M604457200) to monitor DDK subunits in HeLa cells, which would presumably have been helpful here in MCF10A cells. Nevertheless, the DNA sequence of the edited clones indicates frameshift mutations that lead to premature STOP codons, and the various phenotypes reported in this manuscript are consistent with loss of DBF4 / DRF1 function as described.

> We thank the reviewer for their time and effort in carefully assessing the manuscript, and with their positive assessment.

We have now included experimental evidence indicating that DBF4 expression is deficient in the DBF4 KO cells used in this study and that the interaction with DRF1 and CDC7 is deficient in the DRF1-KO cells using the same Co-IP strategy previously reported in HeLa cells. Please see also the response to reviewer #2 to the same point.

Minor comments:

1. The authors should discuss their data in the context of the recent study by Suski et al (<https://doi.org/10.1038/s41586-022-04698>). The latter study reported that knockout of DBF4 in mouse fibroblasts impairs proliferation but is not lethal, in agreement with the present manuscript, but Suski et al also argue that CDC7 is dispensable for DNA replication in mammalian cells due to redundancy with CDK1.

> The requirement for CDC7 kinase activity for genome duplication in mammalian cells has become a contentious point of debate. CRISPR screens in more than 1000 cell lines indicate that CDC7 is a core essential gene required for proliferation (DepMap.org) and we have

previously shown that CDC7 is an essential gene and that the expression of a kinase dead mutant is not able to rescue viability of in MCF10A cells (Rainey et al 2017 <https://doi.org/10.1021/acscchembio.7b00117>).

Suski et al argue that CDC7 is dispensable for DNA replication in human cells based on acute degradation of CDC7 or by its inhibition using an “Shokat type” analogue sensitive CDC7 allele. However, another study showed that DNA replication is not completed using the same approach and the same analogue sensitive allele (Jones et al. 2021 <https://doi.org/10.1016/j.molcel.2021.01.004>)

In mouse embryonic stem cells, the Masai group had previously shown that CRE-Lox mediated inactivation of mDBF4 leads to a strong decrease of DNA synthesis and that mDBF4, like mCDC7 is essential for cell ES cells viability (Kim et al, 2002 <https://doi.org/10.1093/emboj/21.9.2168>) and Yamashita 2005 <https://doi.org/10.1111/j.1365-2443.2005.00857.x>). Intriguingly mDRF1 has yet not been identified nor characterised and might have evolved as a pseudogene. So, it is possible that in mouse that only DBF4 is expressed.

In our view, the conflicting results about the essentiality of CDC7 can be reconciled by acknowledging that human CDC7s is indeed essential, and that the total loss of kinase function is required to block DNA replication, which may have not been fully achieved in the Suski’s study. Incomplete CDC7 inhibition, still allows genome duplication because of the large abundance and redundancy of replication origins which means that activation of only a small fraction of these is truly necessary for full genome replication, and similar observations were previously reported when MCM protein was depleted (Ge et al. 2007 <https://doi.org/10.1101/gad.457807> ;Ibarra et al 2007 <https://doi.org/10.1073/pnas.0803978105>).

Furthermore, several proteins contribute both positively and negatively to the effectiveness of CDC7 inhibition in DNA replication and cell proliferation e.g. RIF1 depletion, ATR inhibition, PTBP1 mutation. (Hiraga et al. 2017 <https://doi.org/10.15252/embr.201641983>; Rainey et al. 2020 <https://doi.org/10.1016/j.celrep.2020.108096> ; Jones et al. 2021 <https://doi.org/10.1016/j.molcel.2021.01.004> ; Göder et al. 2023 <https://doi.org/10.1016/j.isci.2023.106951>).

Specifically, CDK1-phosphorylation of RIF1 was shown to disrupt RIF1/PP1 interaction and PP1’s ability to counteract CDC7-dependent phosphorylation of the MCM complex (Moiseeva et al. 2019 <https://doi.org/10.1073/pnas.1903418116> ; Jones et al. 2021 <https://doi.org/10.1016/j.molcel.2021.01.004>). Thus increased CDK1 activity can be helpful in dealing with low levels of CDC7 kinase, as also we had previously shown (Rainey et al 2017 <https://doi.org/10.1021/acscchembio.7b00117>)

We have revised the text to include most of the above considerations, within the limit of space allowed.

2. Some discussion of the increased frequency of micronuclei in DRF1-deficient cells

compared to DBF4-deficient lines would be useful (c.f. Figure 1F-G).

> In the revised manuscript we have included some discussion on the increased frequency of micronuclei as follow:

“CDC7 inhibition is associated with irregular progression through mitosis, often resulting in the formation of micronucleated cells (Cazzaniga et al.; Martin et al., 2022). Interestingly, while we did not observe a significant change in the percentage of micronucleated cells in DBF4-deficient cells, while these clearly accumulated in DRF1-deficient cells (Figure 1F-G), which could be due to minor impairment of the in DNA replication/repair or to defective chromosome segregation. A tempting hypothesis is that DRF1 may modulate the timing of abscission at end of the cell cycle, a non-essential process in which CDC7 was recently involved and that, if impaired, can lead to micronucleated cells (Luessing et al., 2022).”

3. It would be helpful to present actual p values in Figure 2, rather than asterisks.

> Asterisks report the range in which the p values fall into, which is specified in the legends (**p<0.01, ****p<0.0001) together with statistical test used. We have not changed the approach as we feel that this would make a less crowded figure conveying the same information.

Reviewer #3 (Significance (Required)):

The main strength of this manuscript is the exploration of the division of labour between DBF4 and DRF1 in human cells, regarding the roles of CDC7 kinase during DNA replication initiation, fork progression and checkpoint control. A limitation would be the failure to monitor the level of DBF4 and DRF1 in the CRISPR-edited cell lines, whilst it is also possible that the relative roles of DBF4 and DRF1 might vary in different cell types.

Previous studies of DNA replication in *Xenopus* egg extracts (e.g. Takahashi et al, 2005: doi: 10.1101/gad.1339805) indicated that DRF1 is the dominant activator of CDC7. In contrast, past work from the current authors (Tenca et al, 2007, 10.1074/jbc.M604457200) indicated that DBF4 is the major partner of CDC7 in human HeLa cells, at least at the level of promoting MCM2 phosphorylation (the only parameter monitored in the previous study, whereas the present manuscript goes much deeper into the various roles of CDC7 in DNA replication control and focusses on the role of CDC7 at replication forks and in checkpoint control).

This study should be of interest to those studying chromosome replication, checkpoints and genome integrity. It should also interest those with a more clinical perspective, due to the potential importance of CDC7 kinase inhibitors as anti-cancer agents.

My own expertise is in the field of chromosome replication.

April 29, 2024

RE: JCB Manuscript #202402144R

Prof. Corrado Santocanale
Ollscoil na Gaillimhe - University of Galway
Centre for Chromosome Biology
Biosciences Building
Newcastle Road
Galway H91W2TY
Ireland

Dear Prof. Santocanale,

Thank you for submitting your revised manuscript entitled "DBF4, not DRF1, is the crucial regulator of CDC7 kinase at replication forks." The paper has now been re-assessed by the original referees from Review Commons. We would be happy to publish your paper in JCB pending final revisions necessary to meet our formatting guidelines (see details below).

A. MANUSCRIPT ORGANIZATION AND FORMATTING:

1) Text limits: Character count for Reports is < 20,000, not including spaces. Count includes title page, abstract, introduction, results & discussion, and acknowledgments. Count does not include materials and methods, figure legends, references, tables, or supplemental legends.

2) Figure formatting: Reports may have up to 5 main text figures. Scale bars must be present on all microscopy images, including inset magnifications. Molecular weight or nucleic acid size markers must be included on all gel electrophoresis. Please add scale bars to Figure 2C and MW markers to total protein stain (TPS) images in Figures 1D/E, 4A/B, & 5A.

Also, please avoid pairing red and green for images and graphs to ensure legibility for color-blind readers. If red and green are paired for images, please ensure that the particular red and green hues used in micrographs are distinctive with any of the colorblind types. If not, please modify colors accordingly or provide separate images of the individual channels.

3) Statistical analysis: Error bars on graphic representations of numerical data must be clearly described in the figure legend. The number of independent data points (n) represented in a graph must be indicated in the legend. Please, indicate whether 'n' refers to technical or biological replicates (i.e. number of analyzed cells, samples or animals, number of independent experiments). If independent experiments with multiple biological replicates have been performed, we recommend using distribution-reproducibility SuperPlots (please see Lord et al., JCB 2020) to better display the distribution of the entire dataset, and report statistics (such as means, error bars, and P values) that address the reproducibility of the findings.

Statistical methods should be explained in full in the materials and methods. For figures presenting pooled data the statistical measure should be defined in the figure legends. Please also be sure to indicate the statistical tests used in each of your experiments (both in the figure legend itself and in a separate methods section) as well as the parameters of the test (for example, if you ran a t-test, please indicate if it was one- or two-sided, etc.). Also, if you used parametric tests, please indicate if the data distribution was tested for normality (and if so, how). If not, you must state something to the effect that "Data distribution was assumed to be normal but this was not formally tested."

4) Materials and methods: Should be comprehensive and not simply reference a previous publication for details on how an experiment was performed. Please provide full descriptions (at least in brief) in the text for readers who may not have access to referenced manuscripts. The text should not refer to methods "...as previously described." Please also indicate the acquisition and quantification methods for immunoblotting.

5) For all cell lines, vectors, constructs/cDNAs, etc. - all genetic material: please include database / vendor ID (e.g., Addgene, ATCC, etc.) or if unavailable, please briefly describe their basic genetic features, even if described in other published work or gifted to you by other investigators (and provide references where appropriate). Please be sure to provide the sequences for all of your oligos: primers, si/shRNA, RNAi, gRNAs, etc. in the materials and methods. You must also indicate in the methods the source, species, and catalog numbers/vendor identifiers (where appropriate) for all of your antibodies, including secondary. If

antibodies are not commercial, please add a reference citation if possible.

6) Microscope image acquisition: The following information must be provided about the acquisition and processing of images:

- a. Make and model of microscope
- b. Type, magnification, and numerical aperture of the objective lenses
- c. Temperature
- d. Imaging medium
- e. Fluorochromes
- f. Camera make and model
- g. Acquisition software
- h. Any software used for image processing subsequent to data acquisition. Please include details and types of operations involved (e.g., type of deconvolution, 3D reconstitutions, surface or volume rendering, gamma adjustments, etc.).

7) References: There is no limit to the number of references cited in a manuscript. References should be cited parenthetically in the text by author and year of publication. Abbreviate the names of journals according to PubMed.

8) Supplemental materials: Reports may have up to 5 supplemental figures and 10 videos. Please also note that tables, like figures, should be provided as individual, editable files. A summary of all supplemental material should appear at the end of the Materials and methods section. Please include one brief sentence per item.

9) eTOC summary: A ~40-50 word summary that describes the context and significance of the findings for a general readership should be included on the title page. The statement should be written in the present tense and refer to the work in the third person. It should begin with "First author name(s) et al..." to match our preferred style.

10) Conflict of interest statement: JCB requires inclusion of a statement in the acknowledgements regarding competing financial interests. If no competing financial interests exist, please include the following statement: "The authors declare no competing financial interests." If competing interests are declared, please follow your statement of these competing interests with the following statement: "The authors declare no further competing financial interests."

11) A separate author contribution section is required following the Acknowledgments in all research manuscripts. All authors should be mentioned and designated by their first and middle initials and full surnames. We encourage use of the CRediT nomenclature (<https://casrai.org/credit/>).

12) ORCID IDs: ORCID IDs are unique identifiers allowing researchers to create a record of their various scholarly contributions in a single place. Please note that ORCID IDs are required for all authors. At resubmission of your final files, please be sure to provide your ORCID ID and those of all co-authors.

13) JCB requires authors to submit Source Data used to generate figures containing gels and Western blots with all revised manuscripts. This Source Data consists of fully uncropped and unprocessed images for each gel/blot displayed in the main and supplemental figures. Since your paper includes cropped gel and/or blot images, please be sure to provide one Source Data file for each figure that contains gels and/or blots along with your revised manuscript files. File names for Source Data figures should be alphanumeric without any spaces or special characters (i.e., SourceDataF#, where F# refers to the associated main figure number or SourceDataFS# for those associated with Supplementary figures). The lanes of the gels/blots should be labeled as they are in the associated figure, the place where cropping was applied should be marked (with a box), and molecular weight/size standards should be labeled wherever possible. Source Data files will be directly linked to specific figures in the published article.

14) Journal of Cell Biology now requires a data availability statement for all research article submissions. These statements will be published in the article directly above the Acknowledgments. The statement should address all data underlying the research presented in the manuscript. Please visit the JCB instructions for authors for guidelines and examples of statements at (<https://rupress.org/jcb/pages/editorial-policies#data-availability-statement>).

B. FINAL FILES:

-- High-resolution figure and MP4 video files: See our detailed guidelines for preparing your production-ready images,

<https://jcb.rupress.org/fig-vid-guidelines>.

Thank you for your attention to these final processing requirements. Please revise and format the manuscript and upload materials within 7 days. If you need an extension for whatever reason, please let us know and we can work with you to determine a suitable revision period.

Thank you for this interesting contribution, we look forward to publishing your paper in Journal of Cell Biology.

Sincerely,

Agata Smogorzewska, MD, PhD
Monitoring Editor
Journal of Cell Biology

Dan Simon, PhD
Scientific Editor
Journal of Cell Biology

Reviewer #1 (Comments to the Authors (Required)):

DBF4, not DRF1, is the crucial regulator of CDC7 kinase at 2 replication forks. The authors have generated DBF4-deficient cells and show altered replication efficiency, partial deficiency in MCM helicase phosphorylation and alterations in the replication timing of discrete genomic regions in these cells. CDC7 function at replication forks is entirely dependent on DBF4 and not on DRF1. Thus, DBF4 is the primary regulator of CDC7 activity, mediating most of its functions in unperturbed DNA replication and upon replication interference. The data is compelling and the authors have addressed previous concerns, confusion. It's a great paper.

Reviewer #2 (Comments to the Authors (Required)):

The authors have satisfactorily addressed my major concern by experimentally demonstrating the loss of detectable levels of DBF4 and DRF1 proteins in the edited cell lines. They have also satisfactorily responded in the main text and the rebuttal letter the rest of questions. Once resolved these critical points, I strongly support the publication of the study for its relevance in the field of DNA replication and genome integrity.

Reviewer #3 (Comments to the Authors (Required)):

The authors have done a good job of addressing the points I raised previously. In my opinion, the manuscript is now ready for publication in J. Cell Biol.